

# Top-down and bottom-up controls on mountain-hopping erosion: insights from detrital [10]Be and river profile analysis in Central Guatemala

Gilles Y. Brocard[1], Jane K. Willenbring[2], Tristan Salles[3], Michael Cosca[4], Axel Guttiérez-Orrego[5], Noé Cacao-Chiquín[5], Sergio Morán-Ical[5], Christian Teyssier[6]

[1]Archéorient, University of Lyon 2, 69365 Lyon, France
[2] Department of Geological Sciences, Stanford University, Stanford, CA 94305, USA
[3]: School of Geosciences, University of Sydney, Camperdown, Australia
[4]: U.S. Geological Survey, Denver Federal Center, Denver, CO 80225, USA
[5]: Carrera de Geología, Universitad San Carlos de Guatemala, Centro Universitario del Noreste, 16001 Cobán, Guatemala.
[6]: Department of Earth Sciences, University of Minnesota, Minneapolis MN 55455, USA

*Correspondence to*: Gilles Brocard (gilles.brocard@mom.fr)

**Abstract.** The presence of a mountain affects the circulation of water in the atmosphere and over the land surface. These effects are felt over some distance, beyond the extent of the mountain, controlling precipitation delivery and river incision over surrounding landmasses. The emergence of a new mountain range therefore affects the erosion of pre-existing mountains located in close proximity. We document this phenomenon in the mountains of Central Guatemala. The $^{40}$Ar-$^{39}$Ar dating of lava flows shows that two parallel, closely spaced mountain ranges formed during two consecutive pulses of single-stepped uplift, one from 12 to 7 Ma, and the second one since 7 Ma. The distribution of erosion rates derived from the analysis of detrital cosmogenic $^{10}$Be in river sediments shows that the younger range erodes faster (~300 m/My) than the older one (20-150 m/My), and that erosion correlates with the amount of precipitation. Moisture tracking form the Caribbean Sea is intercepted by the younger range, which casts a rain shadow over the older one. The analysis of river long-profiles provides a record of longer-term interactions between the two ranges. The rivers that drain the older range were diverted by the younger range during the early stages of its rise. A few rivers were able to maintain their course across the young range, through profile steepening, but incision completely stalled along their upper reaches, upstream of the younger range. As a result, the older range has been passively uplifted, and entered a phase of a slow topographic decay: pediments have formed along its base, while ancient headward-migrating waves of erosion, located farther up the mountain flanks, have almost stopped migrating. Aridification and cessation of river incision together explain the slowing down of erosion over the older range. They represent top-down and bottom-up processes whereby the younger range controls erosion over the older one. These controls are regarded as instrumental in the nucleation and enlargement of orogenic plateaus forming above continental accretionary wedges.



## 1. Introduction

The relief of mountain ranges affects the circulation of water as moisture in the atmosphere, and along river networks as surface runoff. The influx of atmospheric moisture is intercepted in the form of precipitation, and returned through stream discharge. The intensity of surface runoff and the rate of river incision control the rate of hillslope erosion. The characteristic length of atmospheric circulation cells and of drainage networks is generally larger than the size of the mountains, such that the effects of a mountain on erosion are felt widely beyond the footprint of the mountain. The most common consequence of atmospheric moisture interception is the concentration of rainfall on the windward side of a range, and the development of an extensive rain shadow that extends beyond its lee-side (e.g.Meijers et al., 2018; Galewsky, 2009). Once precipitated, water and eroded sediments generate a return flux that affects river drainage in many ways. These effects are felt not only downstream of a rising mountain range, but also upstream, because downstream changes affect the evolution of upstream reaches (Humphrey and Heller, 1995; Whittaker and Boulton, 2012), and, therefore, ultimately affect the hillslopes connected to the upstream reaches (Harvey, 2002; Mudd and Furbish, 2007). A mountain ranges therefore exerts top-down controls (e.g. precipitation, vegetation cover, weathering, surface runoff, landsliding), and bottom-up controls (upstream-migrating signals of erosion along stream networks) on erosive processes operating over surrounding areas. The rise of a mountain upwind and downstream of an existing mountain occurs typically during the growth and widening of an orogenic accretionary wedge (Garcia-Castellanos, 2007). Through a combination of enhanced tectonic uplift across river courses, and of aridification, the rise of the new mountain can trigger the rerouting, and sometimes the full disintegration of river networks draining preexisting ranges. These processes are regarded as some of the most potent mechanism of formation of orogenic plateaus (Sobel et al., 2003; Garcia-Castellanos, 2007).

Here we document the erosional and topographic hallmark of these processes in Guatemala, across the narrow Central American land bridge. Elongate ranges have risen since Middle Miocene times along the North American-Caribbean plate boundary. The first range to rise is also the closest to the plate boundary. Deep valleys were incised through the range until the late Miocene (Brocard et al., 2011). In the late Miocene, however, uplift broadened, propagating farther away from the plate boundary. The rise of a new mountain range across the path of rivers that drained the preexisting range led to their rapid tectonic defeat, spurring a pulse of drainage rearrangement, and the aridification of the defeated river valleys (Brocard et al., 2011). Drainage rearrangement consisted in a series of river captures, affecting a deeply dissected landscape. Contrary to expectations, river captures did not cause waves of accelerated erosion. Instead, river incision almost completely stalled upstream of the capture sites (Brocard et al., 2011). In this paper we seek to identify the top-down and bottom-up controls on erosion that arrested river incision in the first uplifted range, upstream of the capture sites. To achieve this, we investigate the geomorphic evolution of the first uplifted range, and contrast this evolution with the evolution of the younger range. We investigated in particular the spatial distribution of hillslope erosion rates in these ranges, provided by detrital terrestrial [10]Be concentrations and the characteristics of the long profile of the rivers that drain these ranges today.



New $^{40}$Ar/$^{39}$Ar ages on volcanic rocks are first provided to better constrain the uplift chronology of the studied ranges. The terrestrial $^{10}$Be concentration of river sediments is then compared to the present-day distribution of precipitation to assess the importance of topographically-controlled rainfall on hillslope erosion. River long-profiles are then used to assess the response of rivers to tectonic uplift, and the response time of river incision to tectonic and climatic changes. These data are then used to assess the influence of river incision on hillslope erosion.

## 2. Origin and evolution of the mountain ranges of Central Guatemala

### 2.1. Tectonics and orogenesis

Left-lateral motion along the transform North American-Caribbean plate boundary has generated elongate, E-W striking ranges parallel to the plate boundary (Fig.1). We investigated three of these ranges, namely the Sierra de las Minas (SM), Sierra de Chuacús (SC) and Altos de Cuchumatanes (AC). Seventy million years of left-lateral wrenching have imparted a deeply penetrative, sub-vertical tectonic grain to the lithosphere (Ratschbacher et al., 2009; Ortega-Gutierrez et al., 2004; Ortega-Obregón et al., 2008). Since Eocene times, however, left-lateral motion has been chiefly focused on the Motagua fault, and, to a lesser extent, on the Polochic fault. With >1,100 km of total cumulative displacement, the Motagua fault is the active on land fault with the largest total offset on Earth. The Polochic fault is thought to branch out of the Motagua fault in the east, in Caribbean Sea, before running on land at an average distance of 50 km from the Motagua fault. The total offset of the Polochic fault reaches 125±5 km (Burkart, 1978). Strain across the plate boundary is strongly partitioned: the left-lateral faults accommodate mostly the left-lateral shear, while motion transverse to the plate boundary is accommodated by subparallel faults with dip-slip reverse and normal motion (Authemayou et al., 2011b; Brocard et al., 2012).



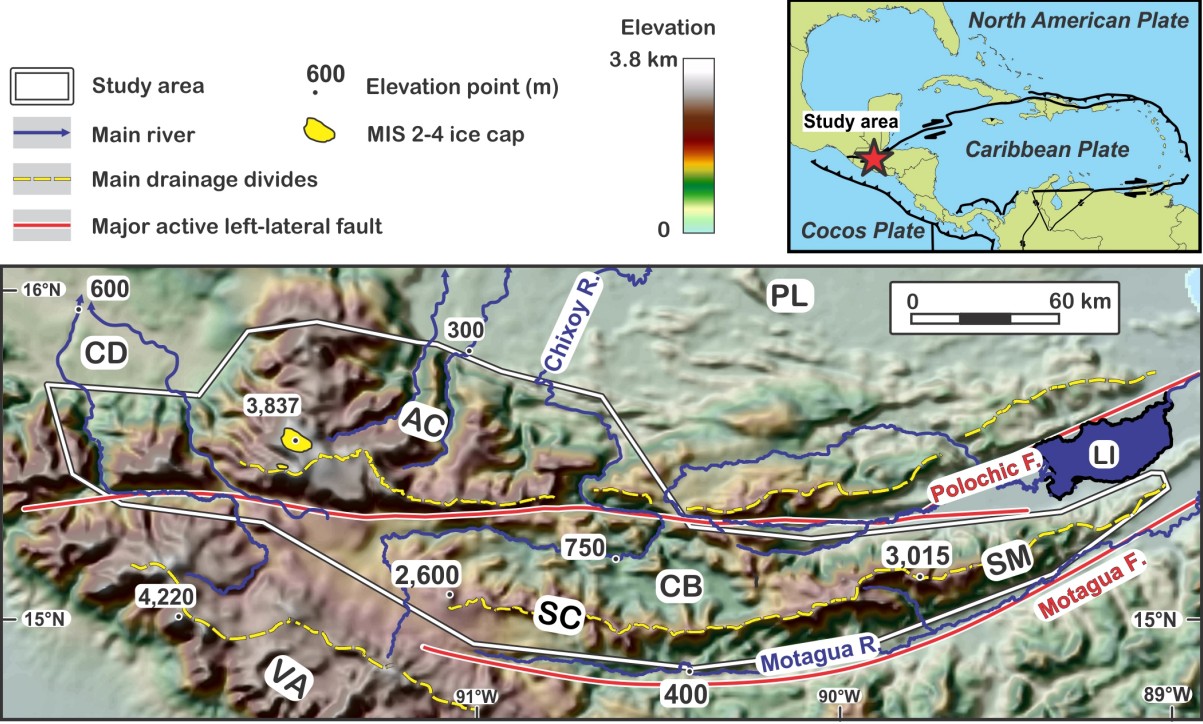

Figure 1. Topographic relief of the study area.

Topographic features: CB: Chixóy River basin, CD: Central Depression of Chiapas, AC: Cuchumatanes Highs, LI: Lake Izabal, PL: Petén lowlands, SC: Sierra de Chuacús, SM: Sierra de las Minas, VA: Central American Volcanic Arc. MIS: $\delta^{18}O$ Marine Isotopic Stage

In Middle Miocene time, the topography of Central Guatemala was very subdued. Remnants of this low relief (referred to as
the Maya surface) still cap many mountaintops across the study area (Fig.2). The low middle Miocene relief formed from the topographic decay of Eocene folds (Authemayou et al., 2011b; Brocard et al., 2011). The mountainous terrain that occupies the study area today started to grow toward the end of the middle Miocene. Surface uplift initially affected both sides of the Motagua fault (Simon-Labric et al., 2013), extending as far north as the Polochic fault (Brocard et al., 2011). This phase of uplift may have been driven by a slab break-off, thought to have occurred below the Central American subduction zone
during the Middle Miocene (Rogers et al., 2002). Valleys up to 1,000 m deep were incised between 12 and 7 My ago into the northern flank of the Sierra de Chuacús (Brocard et al., 2011). 3.1 to 6.1 My-old basalts along the floor of the Motagua valley (Tobisch, 1986) provide further evidence that 1,000-1,500 m of incision also occurred over the period along the Motagua valley (Fig.3).





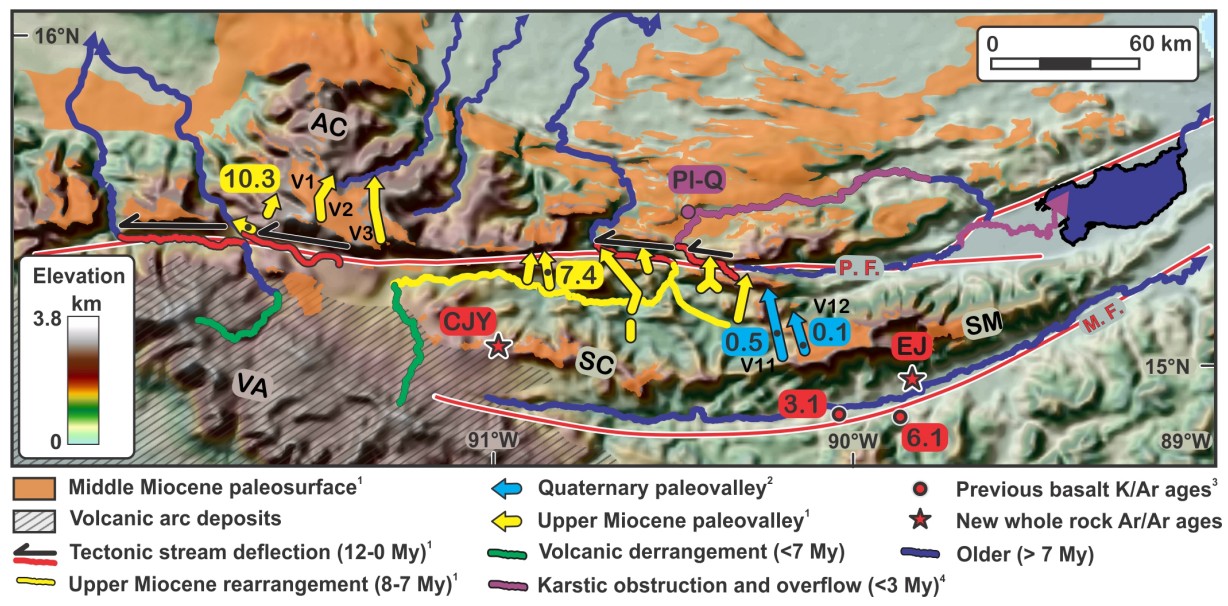

Figure 2. Age of geomorphic markers and drainage lines. Ages of Miocene valleys and Quaternary paleovalleys (V1-V12) provided in My. Data source: 1: (Brocard et al., 2011), 2: (Brocard et al., 2012), 3: (Tobisch, 1986), 4: Plio-Quaternary lacustrine deposits (Brocard et al., 2015a). Newly dated lavas: CJY: Chujuyúb, EJ: El Jute. Range names: AC: Altos de Cuchumatanes, SC: Sierra de Chuacús, SM: Sierra de las Minas, VA: volcanic arc. Faults: MF: Motagua, PF: Polochic. Background: shaded GTOPO 30 DEM.

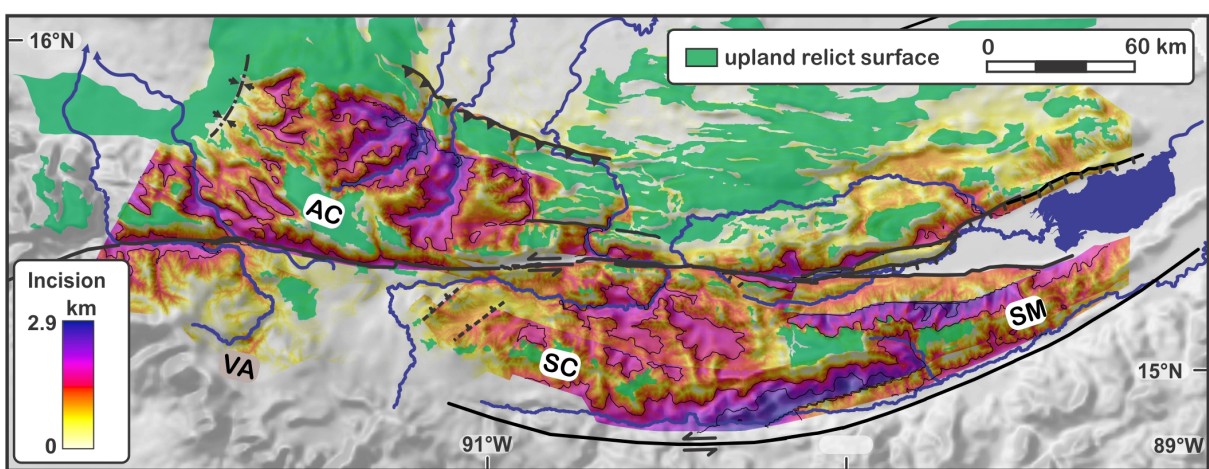

Figure 3. Incision below the Middle Miocene Maya surface, based on the elevation of surface remnants (upland relict surface). Incision contour line spacing: 1km. AC: Altos de Cuchumatanes, SC: Sierra de Chuacús, SM: Sierra de las Minas, VA: volcanic arc.





Contraction north of the Polochic fault (Authemayou et al., 2011a) triggered the rise of the AC range during the Late Miocene (Brocard et al., 2011). A westward decrease in the length of river deflections along the Polochic fault (Fig. 2) may reflect earlier incision of rivers in the west, consecutive to an earlier rise of the AC range in the west. If this

interpretation is correct, it can be inferred, from the length of the tectonic deflections, and from the slip velocity of the Polochic fault, that the AC range started rising in the west ~12 My ago, progressing 150 km eastward during the following 5 My (Brocard et al., 2011). The AC range barred the courses of the rivers that drained the northern flank of the SC-SM range. Many of these rivers were defeated, leaving paleovalleys along the southern flank of the AC range (Fig. 2). The amount of deformation of these paleovalleys indicates that the AC range has risen >1 km relative to the SC range during the past 7 My

(Brocard et al., 2011). Today, the AC range accommodates rock uplift within the hanging wall of the poorly-documented transpressional Ixcán fault, a fault that bounds the range to the north (Guzmán-Speziale, 2010)(Fig. 4). The AC range also accommodates kink folding to the NW (Authemayou et al., 2011b). Surface uplift is fueled by contraction, combined with erosional unloading along the deeply dissected flanks of the range (Fig. 3). A 20 x 30 km ice cap occupied the summit plateau of the AC range (Fig. 1) during the last glaciation (Anderson et al., 1973; Lachniet and Vazquez-Selem, 2005).

Unlike other neotropical mountains (Lachniet and Vazquez-Selem, 2005) the AC range does not preserve the remnants of earlier, more extensive ice caps. Such absence implies that earlier ice caps were smaller, despite climatic conditions everywhere else conducive to more extensive glaciers. The most straightforward explanation of this difference is that the AC range rises fast enough to expose ever increasing areas to snow accumulation, from one glacial cycle to the next.

While contraction prevails in the west of the study area, extension prevails in the east. This difference is due to an

eastward increase in the divergence angle between fault orientation and plate motion (Rogers and Mann, 2007). Transtension between the Polochic and Motagua fault started creating the releasing bend of Lake Izabal during the Middle Miocene, in close proximity to the Caribbean Sea (Carballo-Hernandez et al., 1988). Since then, protracted transtension in the bend has led to the burial of the Maya surface under up to 5 km of terrigenous sediments (Bartole et al., 2019). Transtension at the eastern end of the Motagua fault has, likewise, generated an elongate (125x15 km) fault basin more than 1.4 km deep

(Carballo-Hernandez et al., 1988). Starting in the Pliocene, transtension spread farther west, disrupting the northern flank of the SM range and the western end of the SC range (Authemayou et al., 2011b; Brocard et al., 2012). It continues today at slow slip rates north of the Sierra de las Minas (Brocard et al., 2012), and along the Polochic fault (Authemayou et al., 2012).

Slip on the Motagua fault is purely left-lateral today, but vertical displacements generated narrow sedimentary basins along the fault in the past (Ratschbacher et al., 2009). During Eocene times (Newcomb, 1975) an elongate transform

basin formed north of the fault; it was filled by the fluvial Subinal Formation which has an exposed thickness ≥1,500 m (Hirschman, 1962). The Subinal Fm. was fed by streams draining the northern side of the basin, and by an axial drainage that prefigures the current Motagua River. However the axial drainage may have flowed westward, instead of eastwards, toward



southern Mexico, were Motagua valley-type rock assemblages are found (Abdullin et al., 2016). The Subinal formation is broken into tilted blocks separated by steep angle faults with strike-slip and dip-slip displacements (Muller, 1979; Johnson, 1984). To the north, it seems to lie in tectonic contact against the SC-SM basement rocks, usually along high-angle reverse faults (Muller, 1979; Bosc, 1971a).

## 2.1. Drainage evolution during the rise of the Central Guatemala mountain ranges

The evolution of the drainage network in response to contractional and transtensional uplift has been studied in detail (Brocard et al., 2011; Brocard et al., 2012). The rivers analyzed here correspond to the most stable flow lines of this network. They are deeply entrenched in basement rocks of the SC-SM and AC ranges, and their courses have not drifted since the early stages of mountain growth. At their downstream end, they join river reaches that have experienced dramatic changes since Middle Miocene times (Fig. 2). These changes are the following: fisrt, the rivers that cross the Polochic fault have been deflected by left-lateral slip along the fault, developing tectonic deflections up to 40 km-long (Brocard et al., 2011). Second, the uplift of the AC range defeated range-transverse rivers, promoting drainage reversal along defeated transverse valleys, and the formation of range-parallel streams, such as the Chixóy River (Brocard et al., 2011). Third, transtensional faulting along the northern flank of the SM range has sparked a second pulse of drainage reorganization in Quaternary time, consisting of river captures and avulsions, at the benefits of the Polochic and Chixóy river catchments, and at the expense of the Cahabón River catchment (Brocard et al., 2012). Fourth, the growth of the Central American Volcanic arc further deranged river networks located in the west of the study area. The drainage of the arc is recent and often changing, especially after the many large, caldera-forming eruptions that have reshaped the topography of the arc since the Pliocene (Rose et al., 1999). Fifth, over the karstic highlands, north of the Polochic fault, rapid changes in river courses occur following the opening and closure of subterranean karstic pathways, since at least the Pliocene (Brocard et al., 2015a; Brocard et al., 2016a). All these drainage rearrangements have affected the base levels of the rivers targeted in this study.

## 2.3. Evolution of precipitations

Present-day mean annual precipitation varies sharply across the study area (Fig. 3). Moisture from the Pacific and Atlantic oceans generates orographic precipitation along moisture-facing slopes, while rain shadows develop over inland-facing slopes. As a result, and despite its narrowness (260 km in Central Guatemala), the Central American land bridge hosts a central dry corridor, surrounded by mountain ranges that experience marked asymmetric rainfall. The AC range receives as much as 4-6 m.yr$^{-1}$ of precipitation on its northern flank due to the ingress of Caribbean moisture (Thattai et al., 2003). In the west, the Central American Volcanic Arc intercepts moisture rising from the Pacific Ocean. In the east, the SM range funnels



Earth **Surface**
Dynamics
Discussions

Caribbean moisture along the Izabal Basin, with more precipitation taking place along its northern side, with intense fog interception above 2,000 m (Holder, 2004). The SM uplands are therefore mantled by thick soils, with frequent mass-wasting (Bucknam et al., 2001; Ramos Scharrón et al., 2012). By contrast, the SC range, surrounded on all sides by other topographic barriers, is much drier. Semi-arid conditions are reached on the floor of surrounding valleys (Machorro, 2014). The 7.4 My-old fossil forest of Sicaché (Fig. 2), located in an upper Miocene paleovalley on the southern side of the AC

range, included pine trees and fern trees, growing over thick soils, suggesting mean annual precipitation ranging between 950 and 1,300 mm.yr$^{-1}$. This in an area now dominated by xerophytic vegetation (Brocard et al., 2011). The fossil forest indicate a deeper penetration of the Caribbean moisture in the past, before the abandonment and uplift of the valleys that crossed the AC range.

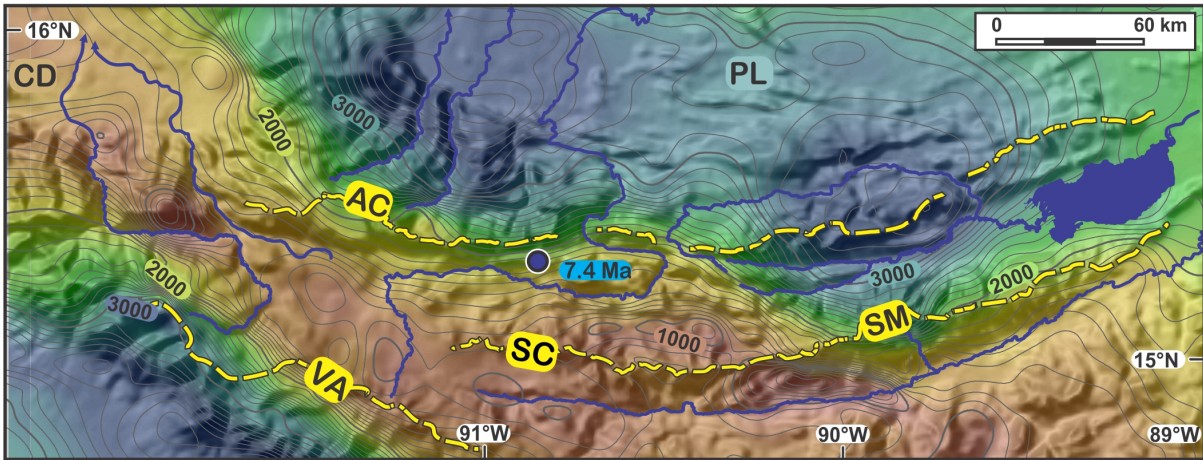


Figure 4. Mean annual precipitation across the study (MARN, 2017) from dry (red) to wet (blue). Isohyet spacing: 100 mm, draped over the shadowed GTOPO 30 DEM. Yellow dashed lines: range drainage divides. Blue dot: location of the fossil forest of Sicaché, buried below a 7.4 My-old ignimbrite (Brocard et al., 2011). CD: Central Depression of Chiapas, AC: Altos de Cuchumatanes, PL: Petén

lowlands, SC: Sierra de Chuacús, SM: Sierra de las Minas, VA: volcanic arc.

### 2.4. Bedrock lithology

Bedrock erodibility exerts a strong control on the response time of river incision to rock uplift (Yanites et al., 2013;

Brocard and Van der Beek, 2006; Finnegan et al., 2005; Duvall et al., 2004). The distribution of rock belts in Central Guatemala tends to follow the strike of mountain ranges (Fig. 5). Schists and gneisses of the Chuacús Fm. form the core of the SC-SM range. They are flanked to the south and to the north by migmatites of the San Agustin Fm., and by marbles and



amphibolites of the Jones Fm. This metamorphic core lies to the north in faulted contact against the basement of North

America, composed of low-grade metamorphic rocks of pre-Permian age (Santa Rosa Fm.) derived from Ordovician-

Carboniferous sedimentary rocks intruded by Ordovician (e.g. Rabinal), Triassic, and Jurassic (e.g. Matanzas) granites. They

underlie the floor of the Chixóy River Basin, between the SC and AC ranges. They also crop out in the core of the AC range.

These basement rocks are overlain by a Permian megasequence of conglomerates, shales, marls, and limestones (Sacapulas,

Tactic-Esperanza, and Chochal Fms. respectively (Anderson et al., 1973)), that mostly exposed in the AC range, and, to a

lesser extent, in the eastern part of the SM range.

A megasequence of continental Jurassic red beds (Todos Santos Fm.), Cretaceous carbonates and Cretaceous

evaporites (Cobán Fm., Campur Fm.) is exposed over much of the AC range (Fig. 5). The strong development of karstic

processes in its carbonate rocks generates complex and rapidly shifting water routing (Brocard et al., 2015a), making the

erosional response of the drainage more difficult to study. For these reasons these areas are excluded from the study.

Ultramafic rocks were obducted over the carbonates in Campanian time. They are preserved within weakly metamorphic

synformal klippes (Baja Verapaz, Santa Cruz, and Juan de Paz). Higher grade serpentine mélanges crop out along the

Motagua valley (Flores et al., 2013).

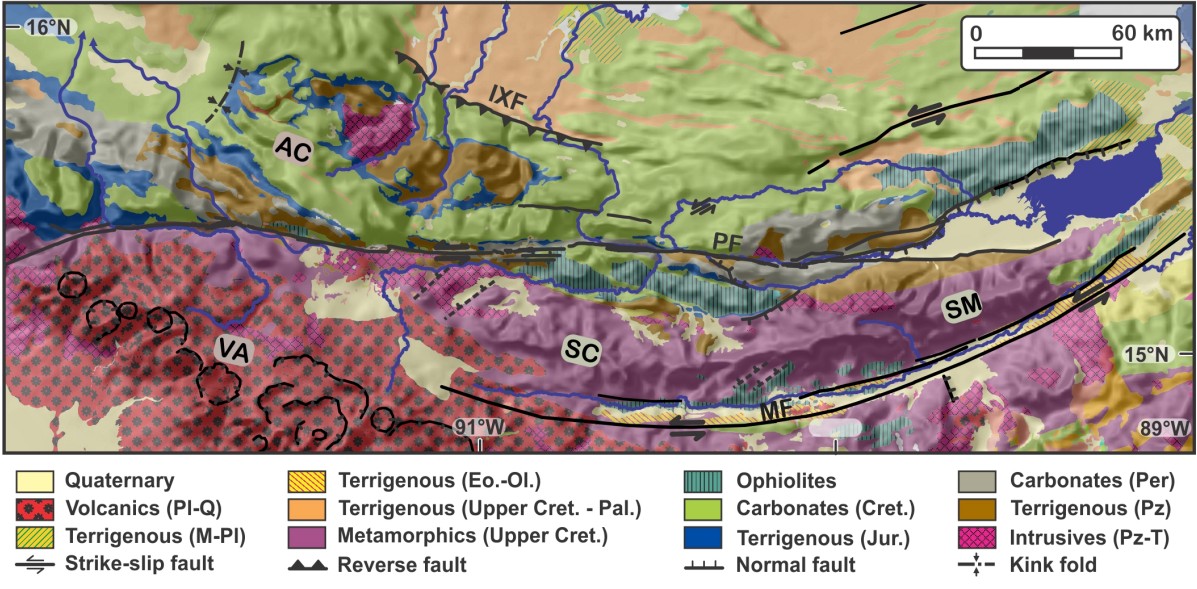

Figure 5. Geology and structure of Central Guatemala (Instituto Geográfico Nacional de Guatemala; Instituto Hondureño de Geología y
Minas; Instituto Nacional de estadística y geografía de México), draped over the GTOPO 30 DEM. Faults: IXF: Ixcán fault, MF: Motagua
fault, PF: Polochic fault.



## 3. Methods


### 3.1. $^{40}$Ar-$^{39}$Ar radiometric dating of volcanic rocks

The age of the low-relief Maya surface was previously constrained by bracketing marker ages. To improve the dating of the surface, we dated a clast of basaltic andesite embedded within a lahar deposit that rests on the Maya surface, at the western

end of the Sierra de Chuacús (Fig. 2). The age of incision of the Motagua valley has been previously constrained using alkaline basalts exposed along the floor of the Motagua valley (Tobisch, 1986). To refine the chronology of incision of the Motagua valley, we dated a basalt flow located 500 m above the Motagua River near El Jute (Bosc, 1971b), in the foothills of the Sierra de las Minas (Fig. 6). Two $^{40}$Ar/$^{39}$Ar whole-rock ages on the basalt of El Jute and one whole-rock age on the basaltic andesite of Chujuyúb were obtained at the U.S. Geological Survey (USGS) in Denver, CO, USA (Appendix A).


### 3.2. Terrestrial $^{10}$Be erosion rates

The concentration of $^{10}$Be in quartz hosted in ridgeline soils and river sediments provides hillslope erosion rates integrated over the past $10^3$-$10^4$ years (Appendix B). The soils and rocks collected along ridgelines located in the SM range provide

site-specific erosion rates (Table S2-1, Fig. S2-3c). They consist of crushed and amalgamated quartzose vein fragments hosted in weathered orthogneiss (3 samples), and of single blocks of highly weathered pegmatite from the Cerro las Palomas, a monadnock located on the Montaña El Imposible (2 samples).

The majority of the dataset, however, consist of $^{10}$Be concentration measured in riverborne quartz, collected in rivers that drain the SM, SC, and AC ranges. They provide catchment-averaged estimates of soil erosion rate (Brown et al., 1995). In

mountainous regions, the amount of $^{10}$Be produced during the downhill transport of quartz grains and their subsequent transport along rivers is usually small compared to the amount of $^{10}$Be produced during the final exhumation of quartz grains through the topmost 2-3 m of soils and rock below the ground surface. The concentration of $^{10}$Be is therefore expected to directly reflect hillslope erosion.

Quartz grains were extracted from the sand grain-size fraction (250-500 µm) of 30 rivers (Fig. 6, Table S2-2, Fig. S2-3).

Many of the sampled rivers drain similar quartz-bearing formations (such as, in particular, the extensive Chuacús Fm. in the SC-SM range, and the extensive Santa Rosa Fm. in the SC and the AC ranges). $^{10}$Be production increases sharply with elevation. Therefore, systematic altitudinal variations in quartz concentration can affect the calculation erosion rates, if homogeneity is assumed. The distribution of quartz-feeding formations in the sampled catchment does not vary consistently with elevation except in the Altos de Cuchumatanes (Fig. S2-3A), where a sensitivity analysis was conducted: we weighted



the $^{10}$Be production according to back-of-the envelope estimates of the relative quartz concentration of the formation outcropping in each catchment. We found the resulting effect to remain marginal, however (<5%, Table S2-2). Quartz enrichment from the fresh rock to the topsoil is commonly important in tropical mountains soils, where weathering is intense. It leads to an underestimation of erosion rates with increasing weathering intensity. An assessment of its potential amplitude was conducted, using quartz enrichment values measured in topical mountain soils from Puerto Rico (Ferrier et

al., 2010).

The sampling was designed such as to document erosion rates within nested catchments (Fig. 6. Fig. S2-3a,b), in order to capture systematic along-stream variations in erosion rates produced by waves of enhanced stream incision along the drainage network (Willenbring et al., 2013).

Samples were prepared at the $^{10}$Be extraction lab of the Department of Geology and Geophysics at the University of

Minnesota, and at the PennCIL lab of the Earth and Environmental Sciences department at the University of Pennsylvania (Appendix B).

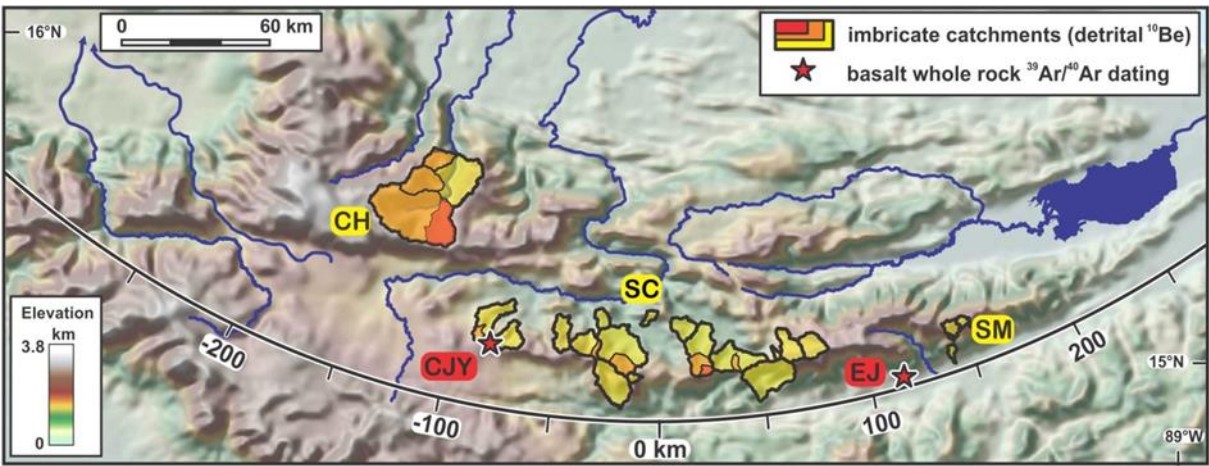

Figure 6. Catchments sampled for the $^{10}$Be analysis (AC: Altos de Cuchumatanes, SC: Sierra de Chuacús, SM: Sierra de las Minas), and $^{40}$Ar/$^{39}$Ar dating (CJY: Chujuyúb, EJ: El Jute). Enlarged maps of the catchments and their lithologies are provided in Fig. S2-1). The arcuate line represents the axis used for plate boundary-parallel projections of Figs. 10 and 12.

### 3.3. Calculation of an erosion index


To test the influence of hillslope steepness and precipitation on $^{10}$Be-derived erosion rates, we calculated a normalized erosion index (Montgomery and Stolar, 2006; Finnegan et al., 2008). The erosion index assumes that hillslope erosion is





driven by surface runoff along hillslopes, during rainfall events. In its simplest form, the entrainment of particles by surface runoff may therefore correlate with hillslope steepness and annual rainfall. The erosion index is calculated as a power

function of these two parameters. Here, it is assumed to be proportional to shear stress: Eq. (1):

$$EI = Q^{1/3}S^{2/3} \tag{1}$$

where Q is the stream discharge (in m$^3$/s) and S the along-stream gradient (m/m). Other choices of exponent values can be made depending of assumptions on the entrainment of soil particles (such as proportional to stream power per unit length or unit area). To assess the sensitivity of the results to the choice of the exponent, we also implemented a more basic version of

the erosion index, in which the index scales linearly with discharge and slope. Slope was extracted from the national Guatemalan IGN DEM at 20 m resolution, and discharge calculated using the mean annual precipitations of the MARN (2017) report. Rainfall values were corrected for evapotranspiration, using a map of vegetation from the (MARN, 2017), and evapotranspiration values representing 10-82% of the total rainfall, depending on the type of vegetation, derived from the Puerto Rico GAP project (Gould et al., 2008). EI values were normalized to the highest obtained erosion index value within

the study area.

### 3. 4. River profile segmentation

To study the response of stream incision to uplift, particularly in the form of upstream-migrating waves of accelerated or

decelerated incision, we extracted the long-profiles of 220 rivers draining the AC, SC, and SM ranges from the Guatemala national 20 m-resolution DEM, released by the National Geographic Institute of Guatemala. Karstic processes, debris flows, and deep-seated landslides dominate the evolution of many small streams, which were therefore excluded from the dataset. A subset of 110 rivers that better capture the evolution of the overall landscape was used in the final analysis (Fig.7).



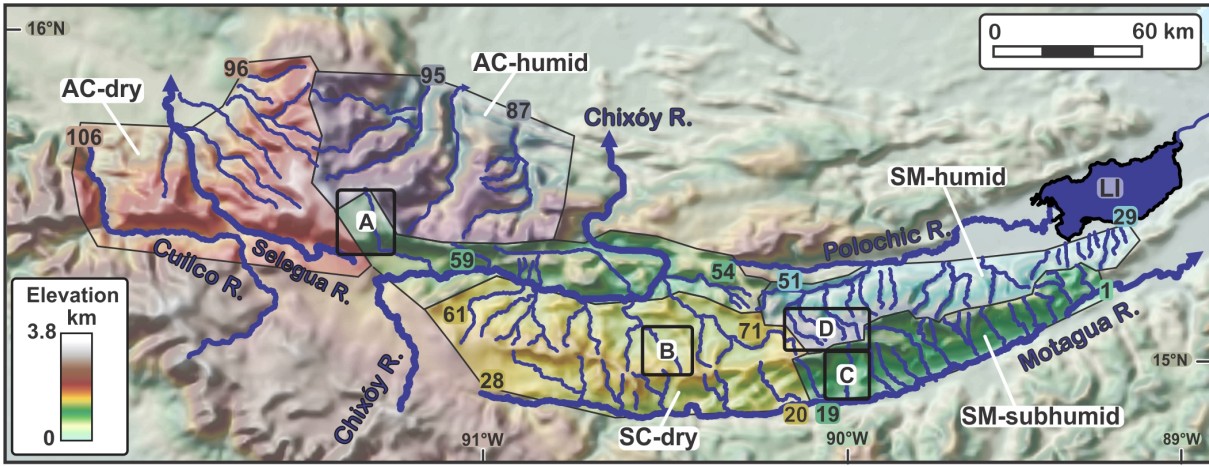


Figure 7. Distribution and grouping of the streams used in the river long-profile analysis, and their grouping by geographic areas. AC: Altos de Cuchumatanes, PF: Polochic fault, SC: Sierra de Chuacús, SM: Sierra de las Minas. Numbers refer to rivers listed in table S4-1. The corresponding profiles are presented on Figs S4-2 to S4-7, and on Fig. 12a,b. Location and extent of the maps displayed on Fig. 8.


River gradients decrease steadily in the downstream direction due to the increase in river discharge. Convex and concave up breaks-in-slope generate irregularities in this overall trend. Convex-up breaks are referred to as river knickpoints. All breaks-in-slope observed here, whether convex or concave-up, are referred to as knickpoints. Knickpoints most commonly manifest adaptations of river gradient to along-stream changes in rock uplift rate, bedrock erodibility, sediment flux, and sediment

grain size. We refer to such knickpoints as steady knickpoints, because their location only changes very slowly, tracking spatio-temporal changes in the distribution of rock types, rock uplift, or sediment fluxes and grain size. By contrast, step changes in base level generate knickpoints located at the front of waves of accelerated or decelerated incision which migrate in the upstream direction. Such knickpoints are here after referred to as migrating knickpoints, as they usually migrate along stream much quicker than steady knickpoints. The identification of knickpoints and their classification as steady or migrating

commonly resorts to a linearization of river profiles that filters out the downstream increase in stream discharge. Among the most commonly used linearization methods one finds the projection of river profiles in DS plots, in which river profiles are represented in a log (downstream distance) vs. local river gradient space (Goldrick and Bishop, 1995), and in AS plots, in which river profiles are projected in a log (local drainage area) vs. log (local slope) space (Howard, 1994). Large step increases in discharge at stream junctions makes linearization results difficult to interpret in AS plot, combined with the

noise commonly observed in slope data, which makes it difficult to separate steady and migrating knickpoints. This is reinforced at places where changes in slope spread over lengths close to the spacing of stream junctions. For this reason, the integral method (Perron and Royden, 2013) has gained popularity. Elevation is plotted (on chi-plots or χ-plots) as a function





of chi (or χ), which is an upstream integral of incremental upstream distance, divided by a normalized local drainage area: Eq. (2):


$$\chi = \int_{xo}^{x} \frac{Ao}{A(x')}^{\frac{m}{n}} dx' \qquad (2)$$

where A and x are the drainage area and upstream distance, respectively; Ao and xo a reference drainage area and a reference upstream distance taken at the same point, respectively, m and n two exponents encapsulating the influence of drainage area and local slope on river incision rate, respectively. One of the caveats of this linearization is that it requires

foreknowledge of ratio the ratio m/n (intrinsic concavity, or θ). The value of θ can be determined by incrementally fitting river profiles to a straight line (Mudd et al., 2014). Such approach requires relatively simple profiles, made up of a small number of successive segments. Many rivers in the study area have highly segmented profiles. The value of θ may also change along-stream, as a result of changes in climate (Murphy et al., 2016), of changes in the dominant erosive processes operating on the streambed (Brocard and Van der Beek, 2006), of the alternation of detachment-limited (Howard, 1994) and

transport-limited river incision (Whipple and Tucker, 2002), and of the alternation of sediment-starved and overfed reaches (Sklar and Dietrich, 2006). Instead of fitting θ to each river profile, we therefore applied a common normalizing concavity θ$_n$ value of 0.5 to all river profiles, after assessing best fit values on a subset of streams (Appendix C). This initial screening showed that most concavity values range between 0.4 and 0.6, as predicted by theoretical studies (Perron and Royden, 2013; Whipple, 2004). Most outliers have higher concavities. Profiles with θ > 0.7 probably incorporate diffuse downstream

changes in dynamics, such as, for example, transitions from boulder-armored reaches to gravel-cobble bars reaches.

## 3.5. Classification of stream segments

The morphology of the streambeds along the linearized segments was determined using stereoscopic black-and-white 0.5-m

resolution aerial photographs shot in 2001 by the Guatemala National Institute of Geography (Appendix D). These observations were punctually ground-proofed in the field (Figs. S4-2 to S4-7). River beds were classified into the following categories, according to the bed component that dominantly controls river gradient: bedrock, bedrock and gravel bars, gravel bars over bedrock strath, gravel bars over thick alluvial fill, colluvium, large immobile boulders, boulders and gravel bars, boulders and bedrock (Table S4-1). Incision is likely detachment-limited along bedload-dominated reaches, and transport-

limited along gravel and cobble-dominated reaches resting over bedrock straths (Tucker and Whipple, 2002; Brocard and Van der Beek, 2006). Identification was left undetermined along many headwater channels, where streambeds are masked by overhanging riparian vegetation.



Changes in streambed type from one segment to the next reflect changes in incision rates, or in bedrock resistance. They therefore assist the identification of the origin of the knickpoints. Boulder-armoured reaches are dominated by slowly-

moving to non-moving boulders that act as a bedrock substrate, rather as a bedload. They are often delivered by valley flanks to river channels and therefore reflect the geology of the hillslopes rather than that of the streambeds. Changes in streambed type are usually sharp along streams across knickpoints. The ranking of segments in discrete classes still somewhat artificially discretizes a rather continuous spectrum of observed streambed conditions when all rivers are analysed together.

**3.6. Classification of river knickpoints**

The knickpoints were classified into the following categories:

1. Lithogenic knickpoints produced by along-stream variations in bedrock erodibility. Downstream increases in erodibility generate concave-up knickpoints, whereas downstream decreases in erodibility generate convex-up knickpoints.

2. Lithogenic knickpoints produced by sudden changes in the relative orientation between the strike of the streambed and the strike and dip/pitch of the dominant structural grain (bedding, tectonic cleavage, foliation, lineation). Erodibility is usually higher parallel to the structural grain. Convex knickpoints develop where stream courses veer from subparallel to crosswise to the structural grain. Conversely, concave knickpoints develop where streams become subparallel to the structural grain.

3. Equilibrium tectonic knickpoints, produced by along-stream variations in rock uplift rate. They are convex-up if rock

uplift and stream incision increase in the downstream direction, and concave-up otherwise.

4. Alluvial knickpoints, generated by downstream changes in grain size. Convex-up alluvial knickpoints result from an increase in bedload grain-size, usually where steep tributaries injects a coarse bedload into a trunk stream. Concave-up knickpoints result from a rapid decrease in grain size, usually through sorting at the boulder-gravel or gravel-sand transitions where bedload grain size distribution is strongly multimodal.

5. Concave-up transitions from detachment-limited to transport-limited river incision, usually at the apex of alluvial fans (Fig. 8c), and pediments (Fig. 8b). Such transitions nucleate at major discontinuities (fault, lithological boundary), and then migrate upstream by backfilling, or by pedimentation. They can be viewed as a concave-up version of the following category (Howard, 1997).

6. Unstable, upstream migrating convex knickpoints, produced by an increase in base-level lowering rate (Rosenbloom and

Anderson, 1994; Merritts et al., 1994). They are convex, separating upstream, slower incising reaches, from downstream, faster incising reaches. Their location is, in principle, independent from lithological boundaries, and controlled by the laws of knickpoint propagation (see below). In some cases, they consist of two successive convexities: an upper convexity set into a soft shallow substrate (alluvium, saprolite), ahead of a second, usually steeper knickpoint, set into the underlying, harder substrate (Fig. 8d). Such knickpoints propagate along many several branches of the drainage network, when produced by a





common change in base-level. In the study area, however, some headward-migrating knickpoint can be tied to more local causes, such as river captures and avulsions (Brocard et al., 2012). A few migrating knickpoints have also formed on resistant rock, upstream of more erodible substrates. Their formation then results from the entrenchment of a previously unsegmented stream into two different lithologies. They are labelled as superimposition transient knickpoints.

7. Miscellaneous knickpoints produced by a variety of other local causes. In the study area, deep-seated landslides entering
river beds have obstructed the rivers and triggered sedimentary back-filling upstream. Steeper profiles are formed across the landslides, where incision rapidly erodes the obstruction itself, or the opposite valley flanks by epigeny. Paroxysmal eruptions along the Central American Volcanic Arc have occasionally caused widespread infilling the valleys of Central Guatemala by volcanoclastic flows. The most recent event of this kind occurred 84 ky ago, during the formation of the Atitlán caldera (Rose et al., 1987), when 100-200 m of primary and reworked pumice filled many valleys (Brocard and
Morán, 2014; Tobisch, 1986). Rivers have long re-incised the fills down to their pre-eruption levels, but one long-lasting effect of this disturbance was the re-incision of some river courses by epigeny away from former valley axes, in the valley flanks, generating knickpoints.

Steady knickpoints correspond to categories 1- 4, while migrating knickpoints correspond to categories 5-6. River profile
analysis have been used to discriminate unstable, migrating knickpoints from stable, equilibrium knickpoints (Goldrick and Bishop, 1995; Perron and Royden, 2013; Whipple and Tucker, 2002). In χ-space, upstream migrating knickpoints which celerity is controlled by the stream power law, propagating along the various branches of a single drainage, and that propagate through a homogenous substrate should share the same elevation and same χ value (Royden and Taylor Perron, 2013). In the real world, variations in bedrock erodibility, climate, and rock uplift scatter these values, challenging
interpretations in particular in areas where heterogeneities generate steady knickpoints with amplitudes and wavelengths similar to that of the migrating knickpoints.

Earth **Surface Dynamics** Discussions
EGU

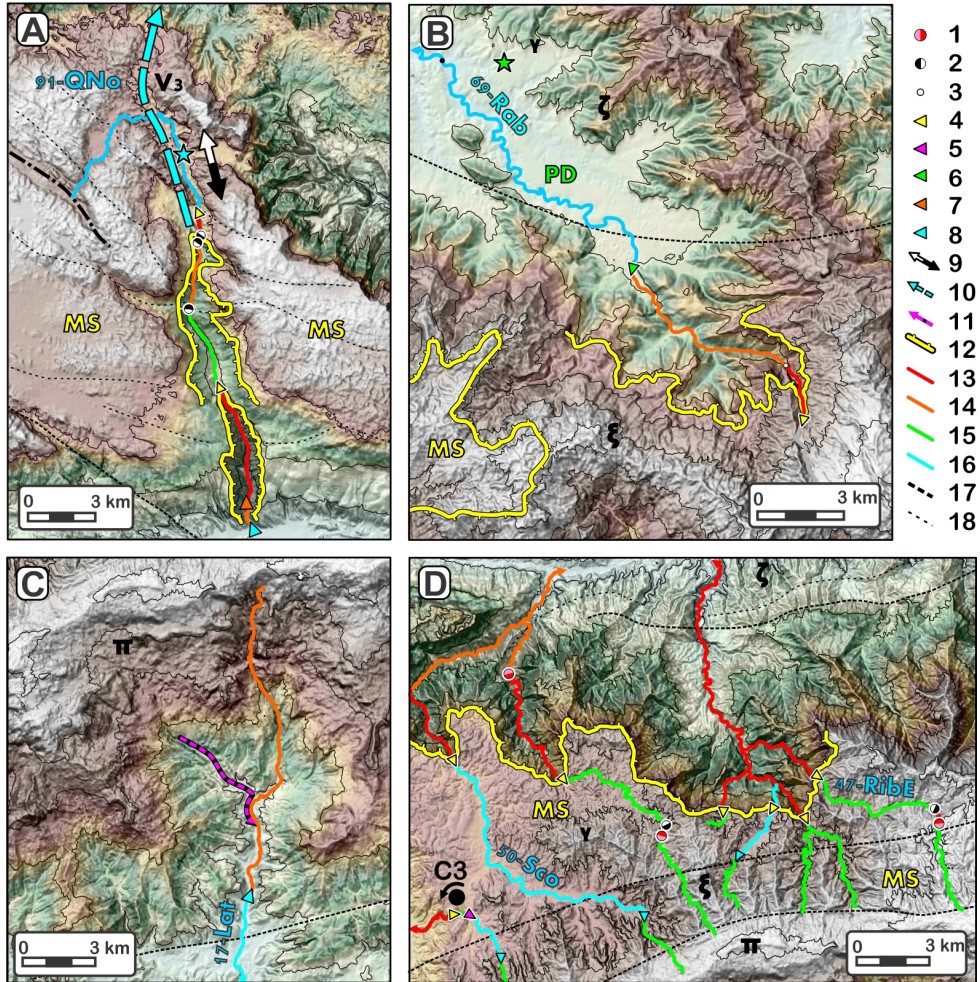

Figure 8. Examples of some knickpoint types in their geomorphic setting. Shaded and sloped 20 m resolution ALOS DEM © JAXA. Location of maps on Fig. 7. (a) paleovalley V3 of Quilén Novillo-Chancol (AC Range), showing the paleovalley shallowly incised in the Maya surface (MS), and its rejuvenation, following drainage reversal, by two nested waves of headward-migrating erosion, the lower one clearly related to the uplift and steepening of the southern flank of the AC range with respect to the SC range farther south. (b): stepped topography in the SC range, with headward migrating knickpoints dismantling the Maya surface, an intermediate wave of dissection, and a pediment (PD) forming at the base of the range, (c): diffusive erosion in serpentinite mélange, catchment of Río Hato (SM Range), (d): dissection of the Maya surface along the northern flank of the SM Range. Blue star: gravel of the QN-Chancol paleovalley (Brocard et al., 2011). Green star: outcrop of pedimented granite.

Lithology: ζ: schist and slate, ξ: gneiss, ϒ: granite, π: serpentinite mélange. Knickpoints: 1: lithogenic, 2: structural, 3: undetermined, 4: headward-migrating, 5: headward-migrating in soft substrate, 6: pediment apex, 7: boulder fan apex, 8: gravelly fan apex, 9: drainage





reversal, 10: paleovalley, 11: large debris flow deposit, 12: front of erosion; river bed environment: 13: bedrock, 14: boulder, 15: gravel over bedrock, 16: gravel, 17: lithological boundary, 18: bedding trace. C3: 200 ky-old avulsion.

Under such circumstances, additional discriminating elements must be used. One consists of checking whether the observed

knickpoints coincide with marked variations in bedrock erodibility or uplift rates, local anomalies, in which case they are likely to be steady, or if, conversely, knickpoints left geomorphic evidence of their migration, or tend to cluster with elevation in which case they are likely to be migrating. We used the 1:50,000 and 1:250,000 geological quadrangles of Guatemala, and topical geologic maps from published papers (e.g. (Brocard et al., 2011; Bosc, 1971a) to assess the effect of lithologic changes on knickpoint location. The stereoscopic black-and white 0.5-m aerial photographs of the Guatemala

National Institute of Geography were used to refine the location of lithological contacts, and to assess the effects of bedrock fabric, fault damage zones, active faults, deep-seated landslides, and large debris flows on the distribution of profile irregularities. We used our foreknowledge of migrating knickpoints produced by Quaternary drainage rearrangement (Brocard et al., 2012), and by active tectonics (Authemayou et al., 2011a; Authemayou et al., 2012; Brocard et al., 2012) to assist the determination of the origin of other knickpoints. Convexities with no obvious local origin were then regarded as

potential headward-migrating knickpoints. The method presents the following limitations: first, local influences can be missed as a result of the imprecision of geologic mapping, especially in the least accessible parts of the SM and AC ranges. Second, large intraformational changes in facies can generate variations in bedrock resistance as sharp as, or even sharper than erodibility differences between mapped geological units. These two effects may lead to the interpretation of stable knickpoints as upstream migrating. Conversely, some headward migrating knickpoints may be pinned to lithological contacts

(Crosby and Whipple, 2006) and filtered out by the analysis. Classification as headward-migrating knickpoint is locally assisted by the presence of break-in-slopes running along valley flanks and tied to a river knickpoints (Fig. 8a,b). They can represent the propagation of the erosive signal upslope, once it has been ensured that such break-in-slopes do not result from lithological variations along the valley sides (which is common in the AC range). Changes in incision rates along hillslopes produced by the passage of a migrating knickpoint can also be marked by changes in drainage density, which reflect changes

in saprolite thickness (Brocard et al., 2015b). In a few cases the passage of migrating knickpoints is marked by the presence of abandoned river terraces and hanging pediments.

## 4. Results

### 4.1. Chronology of base levels lowering from $^{40}$Ar/$^{39}$Ar dating

The Maya surface is thought to have formed close to sea level, because it can be tracked almost uninterrupted from the Caribbean Sea to the Pacific Ocean. It was once covered by extensive fluvial deposits, especially south of the Motagua





fault, where the fluvial deposits are covered by extensive ignimbrites (Williams and McBirney, 1969). The lahar sampled
near Chujuyúb belongs to a sequence of volcanoclastic rocks produced by the Central American volcanic arc and deposited
directly onto the Maya surface. Lahar emplacement predates the incision of a 450m-deep valley. The lahar yielded a plateau
age of 12.54 ± 0.04 My (Fig. S1-1, Table S1-2). Incision at Chujuyúb after 12 My is consistent with the proposed 12 My-old
entrenchment of the Cuilco River valley into the Maya surface, 70 km to the NW (Brocard et al., 2011); with the 10.3 My-
old age of the shallowly-incised Colotenango valley (Authemayou et al., 2012), 35 km to the NNW; and with the > 7.4 My
incision of paleovalleys into the Maya surface, located 10-30 km farther to the NE (Brocard et al., 2011). Considering the
depth of entrenchment of the late Miocene paleovalleys, incision would have proceeded at 140-280 m/My along the now-
abandoned Miocene valleys from 12 to 7 Ma, assuming that the initial dissection of the Maya surface started everywhere at
12 My (a, Fig. 9). Individual valley incision rates, averages over the length of each paleovalley, range between 145 and 205
m/My (b, Fig. 9). These rates stand in sharp contrast with subsequent incision by the Chixóy River and its tributaries which
amounts to not more than a few tens of meters, at rates of <30 m/My, between the SC and AC ranges (c, Fig. 9). The incision
chronology of the northern flank of the Sierra de Chuacús consists of a single pulse of rapid incision from 12 to 7 My, and
then by very slow incision ever since.

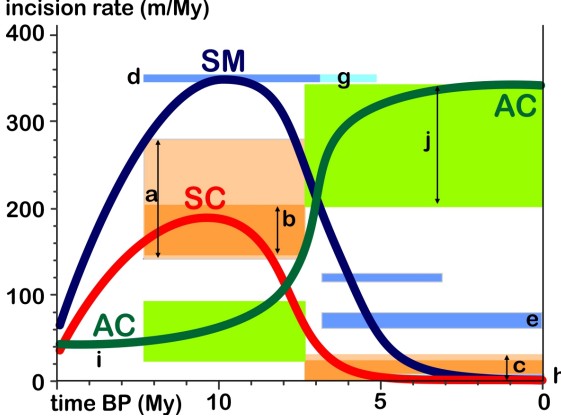


Figure 9. Evolution of incision rates in the studied ranges.

Shades of blue: Sierra de las Minas (SM). Shades of red: Sierra de Chuacús (SC). Shades of green: Altos de Cuchumatanes (AC). a-j: see
main text.

The evolution of the southern flank of the SC range could be more complex due to the presence of a narrow, elongate trough
basin that formed between the range and the Motagua fault during Eocene time. The trough was filled with the continental
red beds of the Subinal Formation. Reactivation of this fault basin in transtension since the Middle Miocene could be



responsible, in part or in whole, for the deepening of the Motagua valley, a possibility contemplated by Tobisch (1986). Various traits of the valley, however, rule out a strong contribution of tectonics to its deepening. First, fluvial sediments

transiting through the valley since the Eocene have bypassed it, feeding a transtensional basin farther east, at the lowest end of the Motagua valley, located beyond the study area. Second, there is no evidence of normal fault slip affecting the alluvial fans that intersect the bounding faults of the basin. The faults encountered along the base of the SC-SM range exhibit ancient, ductile to ductile-brittle left-lateral deformation (Bosc, 1971a; Roper, 1978). Last, the low-relief surfaces that top the summits north and south of the Motagua fault lie at the same elevation (Simon-Labric et al., 2013), implying that if trough

reactivation had occurred, it would be perfectly balanced across bounding faults such that not net tectonic offset is observed from one side of the structure to the other. The deepening of the Motagua valley therefore appears to have been achieved by erosion, through the preferential removal of the erodible sediments filling the fault corridor, giving the valley the appearance of a recently active graben.

Remnants of alkali basalt flows are scattered along the Motagua valley. The lithosphere on the north side of the

valley has not been the source of any Cenozoic magmatism (Simon-Labric et al., 2013). By contrast, the lithosphere on the southern side of the valley is warm, and occupied by a Tertiary-Quaternary back-arc alkaline magmatic province (Walker et al., 2011). The lava flows found on the Motagua valley floor track were all delivered by vents located in the south (Tobisch, 1986). The outcrop of El Jute represents the distal end of a lava flow which, after crossing the Motagua valley, abutted the base of the SM range, backfilling the Huijo River valley with $\geq 70$ m of basalt. The base of the flow lies >400 m above the

axis of the Huijo River valley. Using the modern gradient of the Huijo River valley as a proxy for its 6 My-old gradient, we find that the basalt flow crossed the Motagua River 360 m above the current elevation of the Motagua River. The basalt yields a plateau age of $6.88 \pm 0.03$ My, and a slightly less constrained total age of $6.46 \pm 0.09$ My (Fig. S1-1, Table S1-2). Assuming that the Maya surface started being incised 12 My ago, the 2.6 km deep Motagua valley would have been excavated at ~350 m/My from 12 to 7 My at El jute (d, Fig. 9). Incision would have since proceeded at $79 \pm 4$ m/My. If the

basalt dam was removed rapidly, then incision would have proceeded more slowly, at $59 \pm 9$ m/My. This incision chronology can be refined by adding previously dated basalts (Tobisch, 1986). The closest dated occurrence is the 6.1 My-old Cerro lo de China, located 6 km upstream, along the Motagua River. Cerro lo de China is located on the other side of the plate boundary, such that it was in fact emplaced 120 km farther up the Motagua valley 6.1 My ago. The $3.1 \pm 0.7$ My-old Cerro Onanopa lies 16 km upstream of El Jute, but on the same side of the plate boundary. Its high vesicularity suggests that

it was emplaced at the ground surface, rather than as a shallow sill within the Subinal Fm. Its base lies <10 m above the Motagua River. Strath terraces have been cut by the Motagua River in its flanks (Tobisch, 1986), indicating that it underwent some burial and exhumation. The accordance in elevation between its basal contact and the Motagua River most likely indicates that the Motagua River has only oscillated tightly around its current vertical position over the past 3 My. The incision of the Motagua valley, from the elevation of the El Jute basalt down to that of the current valley floor would thus





have occurred from 6.1 to 3.1 Ma, at > 110 ± 40 m/My (f, Fig. 9). If, after the emplacement of the El Jute basalt, incision continued unabated at ~350 m/My as before the emplacement of the basalt (g, Fig. 9), then the valley would have reached its current depth 5 My ago (h, Fig. 9).

The evolution of incision during the rise of the SC-SM range therefore looks similar on both sides of the range, and is, dominated by a single step of rapid incision at 140-350 m/My from 12 to 7-5 My ago, followed by an almost complete
cessation of incision along the main trunk streams (Motagua and Chixóy River) that represent the base levels of the streams analyzed hereafter.

The late Miocene paleovalleys were deeply incised into the SM-SC range, but only shallowly incised into the AC range (e.g. Fig. 8a; i, Fig. 9), indicating low rates of incision in the AC range, while the SC-SM range were rapidly uplifted. Since then, however, dissection has incised valleys 1,500 m to 2,600 m deep along the northern flank of the AC range,
implying a considerable increase in incision rates as a result of the uplift of the AC range, at 200-350 m/My (j, Fig. 9).

## 4.2. Distribution of $^{10}$Be-derived erosion rates

Catchment-wide detrital $^{10}$Be erosion rates range from 11 m.My$^{-1}$ over the Maya surface in the SM range, up to 330
m.My$^{-1}$ along the wet and steep northern flank of the AC range (Fig. 10). Most slow erosion rates are found in the SC range. Weighting these rates by the relative concentration of quartz in quartz-feeding lithologies only marginally affects these erosion rates (by 3.4% on average in the SC range, 4.8% in the SM range, and < 7% in the AC range). Quartz enrichment in the topsoil could increase erosion rates by up to 40% (Fig. 10, Table S2-2). Quartz enrichment increases with weathering intensity, such as it is probably less pronounced in the AC range, where soils erode the fastest. Quartz enrichment would
therefore tend to reduce the contrast in erosion rates between the SC-SM range and the AC range, without suppressing it.

The nested catchment analysis (arrows, Fig. 10) reveals a marked downstream increase in erosion rate in the AC range (from CATA to CHEL to XAC). In the SC range, a downstream increase is expected in the headwaters, due to a decreasing contribution of slowly-eroding low-relief uplands with downstream distance (Willenbring et al., 2013). Farther downstream, however, a decrease is expected, when the rivers start crossing pediments located on the floor of the Chixóy
River basin. The expected increase is observed (from PAS to PAE), but it is much less pronounced than in the AC range. The expected decrease is also observed (from XEU to CUB), but it is likewise quite subdued. In one case, the expected succession of increase and decrease is not observed (from SMS to SMM to SMI).

In the SM range, catchments are not nested but they still display increasing rates of erosion down the mountain flank, as the entrenchment in the Maya surface increases (COL to FRI to RAN), with one outlier (SLO). The magnitude of
increase is intermediate between those observed in the SC and CA ranges.



Earth **Surface**
**Dynamics**
Discussions

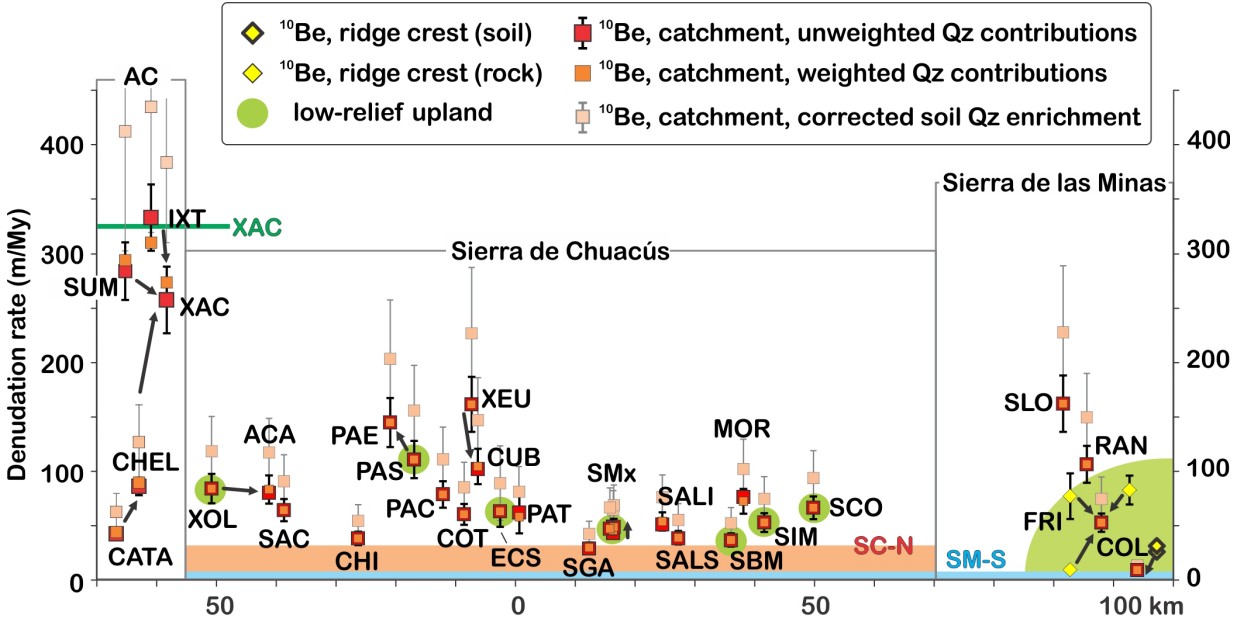

Figure 10. Variations in detrital [10]Be denudation rates along the strike of the plate boundary, from the AC range in the west, to the SM range in the east. Data are projected along the strike of the plate boundary, on an arc displayed on Fig. 6. Arrows shows feeding directions, from hillslope sites to nested catchments. Low relief uplands drain remnants of the Middle Miocene Maya surface. Denudation rates are compared to incision rates along the northern (salmon, SC-N) and southern (blue, SM-S) base of the SC-SM range, over the past 7 My. XAC: peak incision rates in the AC range along Río Xacbal, where incision below the Maya surface along the river course is maximum (Fig. 3).

## 4.3. Distribution of streambed types

The analysis of streambeds was conducted along each straight segment obtained from the linearizing in χ-space. Among the final 93 rivers and 452 segments retained in the analysis, 9% have no knickpoint, 16% host one knickpoint, 51% host 2-5 knickpoints, 25% host 5-12 knickpoints (Fig. 11). Among segments, 92% are linearized, 6% are concave, and 2% are convex.



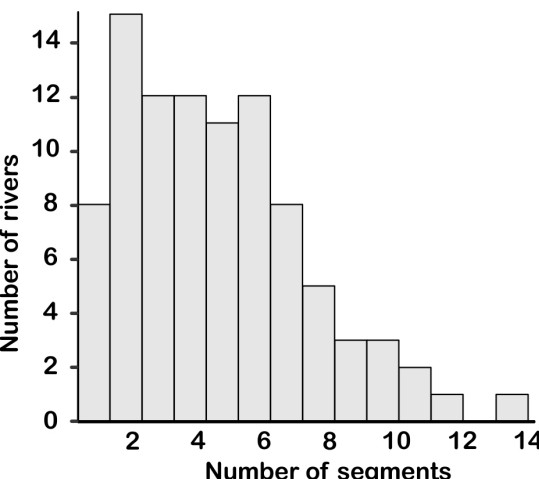

Figure 11. Statistical distribution of the multi-segmented rivers in the studied dataset.

The distribution of alluvial reaches with elevation along the strike of the SC-SM and AC ranges (Fig. 12a1-b2) is bimodal, the majority of alluvial reaches being located either at the base of the mountains (along the Motagua valley, the Chixóy River basin, the basin of Lake Izabal, the Central Depression of Chiapas and the Petén lowlands), or at high elevation, on remnants of the Maya surface and along rivers slightly entrenched into it (e.g. Fig. 8a,d). The alluvial reaches located at high elevation carry a sandy to gravelly bedload, derived from the weathering of underlying crystalline rocks. Gravel is derived

from weathering-resistant quartzose veins and silicified pegmatites (Brocard et al., 2012). Alluvial reaches at intermediate elevations are less common. They form upstream of landslides (notably in the SM-SC range), within localized areas of tectonic subsidence (especially along the Polochic fault corridor, on the southern flank of the AC range), and over extremely erodible lithologies such as fault damage zones and fault gouge.

On crystalline rocks, boulder armoring is more common in the wet parts of the SM range than on the dry parts of

the SC range. In the SM range, many boulder-strewn reaches result from the winnowing of the matrix of debris flows that invaded the streambeds. The SM range is first range hit by Atlantic tropical depressions tracking from the Caribbean Sea. High intensity rainfall events trigger considerable and frequent landsliding along its wettest slopes (Ramos Scharrón et al., 2012; Bucknam et al., 2001). The SM range may also be more susceptible to earthquake-triggered landsliding that the SC range, because it soils are more likely to be water-saturated when an earthquake occurs. Boulder armoring is also common

over serpentinite mélanges along the southern flank of the SM-SC range, owing to the high erodibility of the matrix of the mélanges, and to the presence of large knockers embedded within it (e.g. Fig. 8c). Boulder-strewn reaches are also more frequent than bedrock reaches in the AC range. There, rivers frequently flow over phyllites of the Tactic Fm., but their beds





are armored by boulders shed by hillslopes along which most resistant beds of the Tactic Fm. are exhumed. They are also commonly contributed by the overlying sandstone and limestone of the Todos Santos and Cobán Fms.


## 4.5. Distribution of knickpoint types

About 40% of the 350 identified knickpoints can be tied to variations in bedrock erodibility, while 8% are related to active tectonics, 21% to upstream-migrating waves of accelerated erosion, and 14% to upstream-migrating waves of
decelerated incision. 11% are composite and include some combination of the above.

Concave-up knickpoints fringe the southern base of the SM-SC range (Fig. 12-a1). They are located at the apex of broad, alluvial fans, upstream of the transition between resistant basement rocks and more erodible ophiolitic mélanges or Eocene red beds. They are also found along the northern base of the SM range, at the apex of the alluvial fans that grade to
the fill of the Lake Izabal basin. Along the northern base of the SC range, concave-up knickpoints are located farther into the range, away from the range-parallel trunk steam (the Chixóy River), from which they are separated by broad, pedimented valley floors. These pediments are partially covered by pumice emplaced during the last paroxysmal eruption of the Atitlán caldera 84 ky ago. The pumice has long been dispersed from below the riverbeds, such that the rivers rest directly on the underlying buried pedimented surfaces. In the AC range, most concave-up knickpoints are located above the Ixcán reverse
fault and mark the transition between the AC range and the Petén lowlands.

Lithogenic knickpoints produced by large changes in bedrock resistance or by changes in the orientation of the streams with respect to the fabric of the substrate are frequent halfway down the mountain flanks. Some are also produced by subtle changes in bedrock resistance farther up the mountain flanks, where crystalline rocks weather to sand and gravel, and where streambeds are not armoured by immobile boulders or large cobbles (e.g. Fig. 8c). The absence of boulder armouring
makes such stream more sensitive to bedrock erodibility variations. Similar sensitivity is also found high up the AC range, where limestones and sandstones deliver little sediment to the streams. Another series of lithogenic knickpoints are found in the lowest reaches of streams draining the northern flank of the AC just before they join the Chixóy River, downstream of pedimented valley floors (Fig. 8b). A possible explanation is that the coarsest bedload is retained on the pediments, increasing their sensitivity to changes in bedrock erodibility farther downstream.

The majority of tectonic knickpoints are concentrated on the southern flank of the AC range along a few 100s of meters from the active trace of the Polochic fault, owing to the growth pressure ridges, and within 4 km of the fault owing to the opening of narrow transtensional corridors (Authemayou et al., 2012). No tectonic knickpoints are found along the southern flank of the releasing bend of Lake Izabal, suggesting that no large normal fault is present there. This is consistent with the sedimentary architecture of the basin (Bartole et al., 2019; Carballo-Hernandez et al., 1988) which indicates that



most of the vertical throw over the lifetime of the bend has taken place along the northern side of the basin. Conversely, the overall downstream steepening of river profiles along the northern flank of the AC range over the hanging wall of the Ixcán transpressional fault probably reflects increasing rock uplift within 20 km of the fault. This steepening is commonly rather diffuse, but at places it is associated with a change in bedrock lithology, and expressed by a change in streambed type. Such knickpoints are therefore classified as composite, tectonic-lithogenic knickpoints (Fig. 12b2).

Among the population of headward-migrating knickpoints, a few are associated with well-identified and dated river diversions (S3-1 to S3-3, Fig. 12a2) which occurred during the Quaternary along the Cahabón River (Brocard et al., 2012). Avulsions and river captures sparked the incision of canyons 500 to 1,000 m deep, spearheaded by prominent migrating knickpoints. Most headward-migrating knickpoints are located along the margins of upland low-relief surface remnants (Fig. 12 a1 and a2; Fig. 8d). The fact that they seem less common in the AC range stems from the fact that because large tracts of

the low-relief areas are heavily karstified and lack an organized continuous water routing network, that would otherwise produce such knickpoints as it flows away from these low-relief areas. A few clusters of headward-migrating knickpoints are found halfway down the flanks of the ranges. One such cluster affects the lower southern flank of the SC range. Its elevation coincides with that of tertiary ignimbrites (Fig. 12a1). Another cluster is found half way down the northern flank of the SC range. It is observed in the Chixóy River catchment, but not along the northern flank of the SM range, in the Polochic-

Panimaquito rivers catchment. A poorly defined cluster seems to exist along the southern flank of the AC range. There, headward migrating knickpoints are locally well-expressed (e.g. Fig. 8a), but the ascription of some knickpoints to this cluster is probably obscured by the presence of many tectonic disruptions along the Polochic fault. Another cluster of headward-migrating knickpoints is present halfway down the northwest flank of the AC range, toward the central depression of Chiapas (Fig. 12). There, the horizontal bedding of the bedrock makes it difficult to separate headward-migrating

knickpoints from lithogenic-knickpoints, because they generate similar topographic landforms. Some of the knickpoints identified as lithogenic could be as well headward-migrating knickpoints. This may lead to an underestimation of the number of headward-migrating knickpoints along the northern and northwestern flanks of the AC range.



Figure 12. (a) and (b). Distribution of linearized stream segments and knickpoints along the SC-SM range.





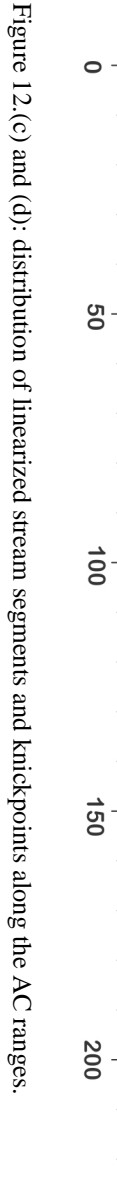

Figure 12. (c) and (d): distribution of linearized stream segments and knickpoints along the AC ranges.




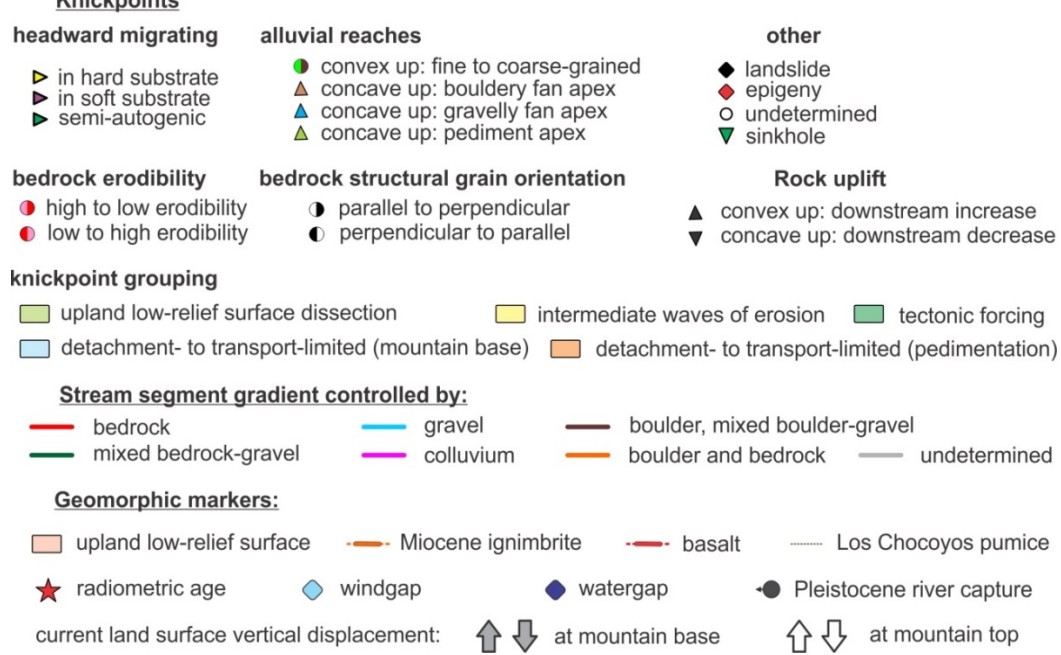

Figure 12. Distribution of linearized stream segments and knickpoints along the SC-SM and AC ranges. Mountain ranges are
projected on the plate boundary, according to a small circle defined on Fig. 6. (a) and (b) southern and northern flanks of the
SC-SM range, (c) and (d): southern and northern flanks of the AC range. Key to abbreviated stream names is provided in
Table S4-1 and in the captions of figures S4-2 to S4-7. Watergap names: CUI: Cuilco, SEL: Selegua, CHX: Chixóy.
Paleovalley numbering from Brocard et al. (2011), river capture numbering from Brocard et al. (2012). LA: city of Los
Amates


## 5. Discussion

### 5.1. Top-down and bottom-up controls on hillslope [10]Be erosion rates

The decrease in erosion rates from the AC to the SC range, and from the SM to the SC range, matches the decrease
in precipitation from north to south, and from east to west generated by the interception of moisture by the AC and SM
ranges and by the development of rain shadows extending over the southern flank of the SM range and over the entire SC





range. As a result, silicate weathering is three times faster along the windward side of the SM range than along its leeward side (McAdams et al., 2015). It is likely, therefore, that precipitation-driven chemical and mechanical weathering play an

important role in the present-day distribution of erosion rates across Central Guatemala. This role is assessed using two calculations of the erosion index: one that assumes that erosion scales with local precipitations and hillslope gradient (as a proxy for soil moisture and chemical weathering), and another that assumes that erosion scales with shear stress during overland flow (as a proxy for mechanical weathering). At first glance, catchment-averaged $^{10}$Be detrital erosion rates scale with precipitation and slope (Fig. 13a,c). The relationship is linear in the AC range, which displays a six-fold increase in

erosion rate, from 50 to 300 m/My, from the interior of the range to its front. This downstream increase results from the concentration of precipitation near the northern front of the range, but it may be combined with an increase in tectonic uplift rate on the hanging wall of the Ixcán fault. The correlation between the erosion index and hillslope erosion rate is also strong from one range to the next, but is weaker within the two other ranges.  t is the weakest in the SC range, where erosion is the slowest (30-70m/My).


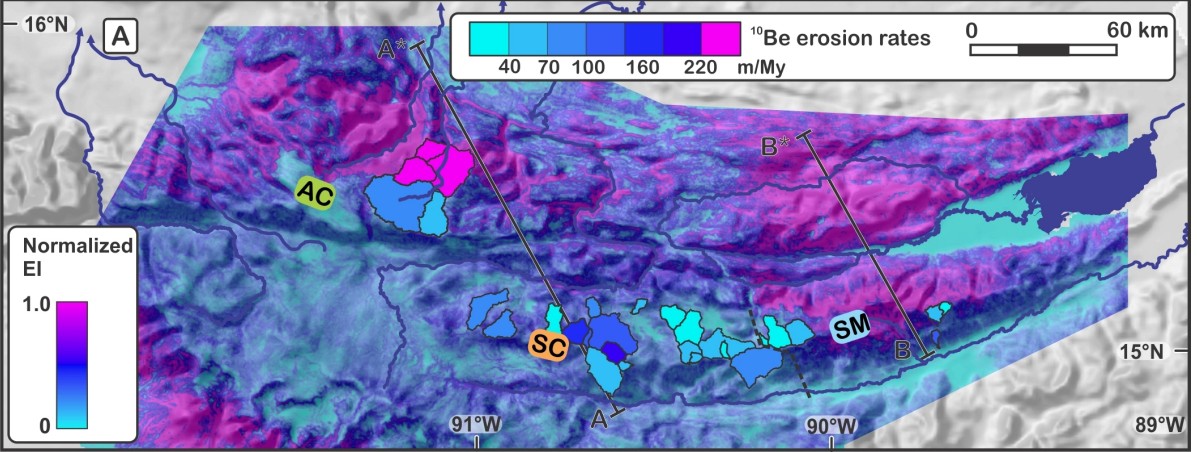

Earth **Surface**
**Dynamics**
Discussions

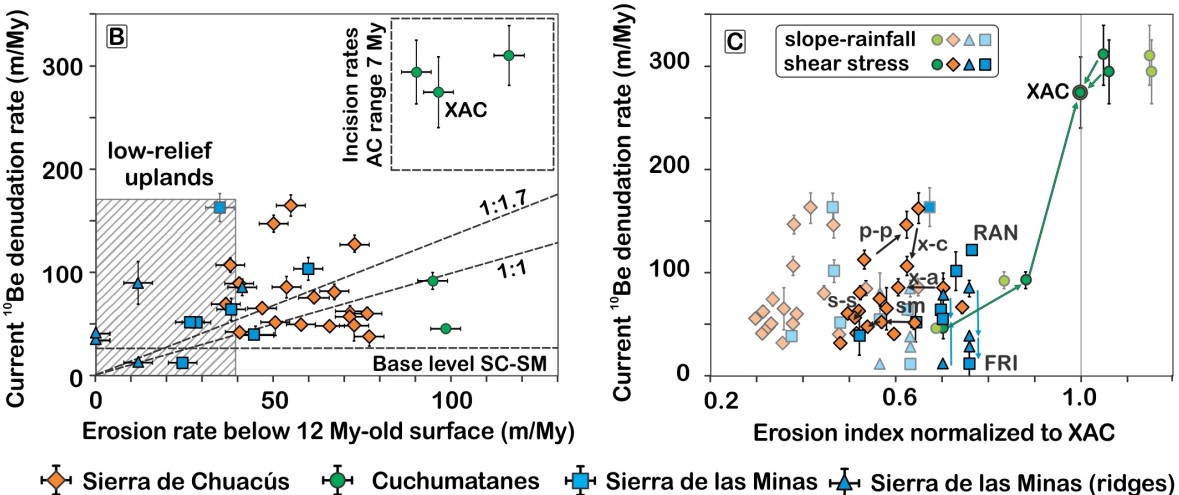

Figure 13. (a) Map showing normalized catchment-wide [10]Be erosion rates superposed to normalized erosion index. (b) and (c) [10]Be catchment-wide erosion rates vs. catchment-averaged dissection depth below the Maya surface (b) and vs. catchment-averaged erosion index (b), for two models of erosion index: slope-rainfall and slope-runoff (shear stress model). Arrows indicate nested catchment links (p-p: PAS to PAE, sm: SMS to SMM to SMI; s-s: SALS to SALI, x-a: XOL to ACA; xc: XEU to CUB).

The downstream increase in erosion rate along the southern flank of the SM range (from FRI to RAN, Fig. 13b) scales with both the increase in EI and with the depth of erosion below the low-relief uplands (Fig. 13b). Because climate is drier down the Motagua valley, this increase in erosion rates is mostly contributed to by the increase in slope. In the SC range, the downstream increase in slope in the nested catchments (p-p) produces a smaller increase in erosion rate despite the same increase is EI. Farther down the range, pediments combine with the dry climate to decrease the average EI of the catchments (x-a, x-c, s-s, and sm, Fig. 13b). Yet, in 3 out of 4 instances, this decrease is not reflected by a decrease in measured hillslope erosion rates. Therefore, it seems that at such low erosion rates, the changes in slope and/or annual precipitation are poor proxies of the processes that drive erosion.

A comparison of current hillslope erosion rates and of long-term hillslope erosion rates (approximated by the amount of dissection below the low-relief uplands averaged over each catchment) provides some insight into the evolution of these erosion rates over the past millions of years: in the fast eroding catchments of the AC range, current hillslope erosion is much faster than dissection below the 12 My-old low-relief upland (line 1:1 on Fig. 13b). It is likely that most of their incision has taken place over the past 7 My (line 1:1.7, Fig. 13 b), because the upper Miocene paleovalleys of the AC range are only slightly incised into its low-relief uplands (e.g. Fig. 12b1). Over the past 7 My, AC range erosion rates are similar to its present-day hillslope erosion rates, at 200-350 m/My (Fig. 9). The overall increase since the upper Miocene is compatible





with the interception of precipitation on the windward flanks of the AC range, and also with the rapid lowering of river base

levels along the northern side of the range, which also promoted valley incision. In the SM and SC ranges, the current rates

of hillslope erosion are comparable to dissection rates below the low-relief uplands integrated over the past 12 Ma. The four

steepest catchments of these ranges do not include low-relief uplands or pediments. In these catchments, present-day rates

are 2-3 times higher than long-term dissection rates. Considering that a most of the dissection occurred before 7 My ago, it is

likely that these modern erosion rates are significantly higher ($> 50$ m/My) than the $<30$ m/My base level fall of the streams

that drain the SC range over the past 7 My. Having higher hillslope erosion rates than valley incision rates means that the SC

range and the southern flank of the SM range have entered a stage of long-term topographic decay. While the increase in

hillslope erosion rate in the AC range may combine an increase in precipitation with an increase in base level lowering rate,

the topographic decay of the SM range seems to combine a decrease in base lowering rate to a decrease in precipitation. Both

act as to lower hillslope erosion rates, but they also decrease the rapidity of the topographic decay of the SM range. The

origin of the decrease in valley incision rates is analyzed in the following section.

**5.2. Bottom-up control on river incision**

**5.2.1. Tectonic control of river incision rate**

**5.2.1.1. Channel steepness**


The projection of river profiles in $\chi$ space (Figs. S4-2 to S4-7) shows that, in each range, linearized segments with

similar streambed conditions (alluvial, boulder-armoured, bedrock) share similar $\theta_n$-normalized steepness values (Fig. 14a).

Rivers flowing over bedrock are, in most cases, steeper than rivers flowing over immobile boulders, which in turn are steeper

than streambeds displaying alternations of gravel bars, bedrock and boulder, which, in turn, are steeper than entirely alluvial

reaches. Among each category, however, differences are observed from one range to the next (Fig. 14b,c,d).

Bedrock reaches are detachment-limited; their gradient are therefore expected to conform to the predictions of the

stream power law (Whipple and Tucker, 1999). At dynamic equilibrium, the steepness of such channels is a function of

streambed erodibility, stream discharge, and rock uplift. The progressive increase in steepness from the SC range to the

southern side of the SM range, to the northern side of the SM range, and finally to the AC range, does not result from an

increase in bedrock erodibility, as erodibility is roughly similar along the strike SC-SM range, and higher in the AC range.

Therefore, the steeper reaches of the AC range and of the northern SM range are better explained by faster river incision.

This is consistent with higher [10]Be hillslope erosion rates in the AC range. Higher precipitation along the N and NW flank of





the AC range, and along the N flank of the SM range does not modify the intrinsic concavity of river profiles enough to prevent linearization for $\theta_n=0.5$.

715       The changes in the steepness of bedrock channels is mimicked by the changes in steepness of boulder-armoured channels and by those of bedload-dominated channels (Fig. 14b,c). While boulders may behave like bedrock, bedload channels are likely transport-limited. In both cases, however, the increase most likely reflects an increase in boulder and bedload grain size. The increase in grain size with increasing incision rate would result from a more limited comminution of bedrock blocks in hillslope soils, reflecting their shorter residence time in saprolites and soils. The effect is stronger among

boulder reaches, and may result from two processes. First, higher intensity rainfall events in the SM range (Bucknam et al., 2001) and in the AC range relative to the SC range may allow them to straddle the transition from diffusive-driven hillslope erosion to landslide-driven erosion. The abundance of debris flows in the SM range supports this possibility: in Panama, the transition to landslide-driven erosion occurs at mean annual rainfall values of 800-1,000 mm (Stallard and Kinner, 2005), which is the threshold crossed when moving from the SC to SM range. Second, in the AC range, where fewer debris flow

deposits are observed, coarsening may be result from the delivery of large blocks from the resistant formations that cap the upper slopes, rather than from the increase in incision rate alone. The specific contribution of tectonic uplift to river steepening in the AC range is further explored in the following section.



Earth **Surface**
**Dynamics**
Discussions

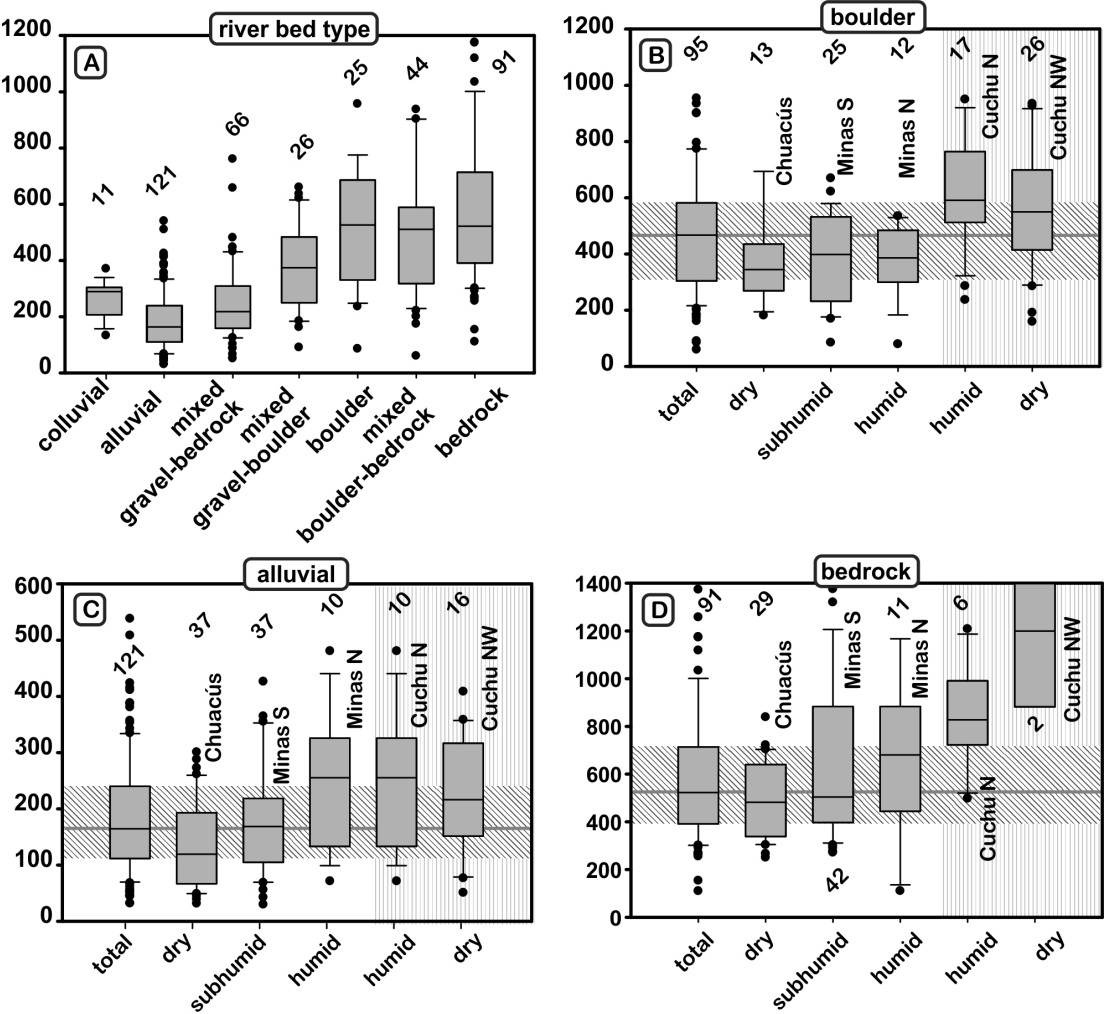

Figure 14. Box plots of stream segment normalized steepness as a function of streambed type and location.

(a): Comparison of steepness by streambed environment; (b), (c), and (c): spatial variations in channel steepness across the study area, according to the regions defined on Fig.7, for the three main types of streambed environment: (b): boulder, (c): alluvial, (d): bedrock. Numbers above box plots: number of segments. Oblique hatches: mean defined by the total number of segments. Vertical hatches: actively uplifting AC range.


### 5.2.1.2. Tectonic steepening across the AC range

Uplift in the AC range led to the diversion of many rivers. Among the three rivers were able to maintain their course across the range, Two (the Chixóy and Selegua rivers) exhibit significantly steeper profiles as they cross the AC range (Fig. 15).



The lithologies incised by these rivers across the AC range are also crosses by the same rivers farther upstream, where they do not generate similar (drainage area-normalized) steepening. Considering that increased discharge would decrease river gradient, rather than increase it, we conclude that the increase observed across the AC is a response to faster tectonic uplift of the AC range (Leland et al., 1998; Kirby, 2003). The lack of steepening along the third river (the Cuilco River) may imply that the range is no longer uplifting in the west. This is at odds, however, with ongoing contractional deformation

documented nearby, at the western end of the Polochic fault (Authemayou et al., 2012). Alternately, it may reflect the fact that the Cuilco River is alluviated over its entire length, and therefore transport-limited. Transport-limited conditions generate less steepening than detachment-limited reaches in response to enhanced rock uplift (Cowie et al., 2008). Transport-limited conditions along the Cuilco River, in turn, result from the large bedload production of the Central American volcanic arc, located nearby in the headwaters of the Cuilco River. Similarly, the Chixóy River collects gravel in its headwaters

within the volcanic arc. This bedload, however, is delivered to the river farther upstream. By the time the Chixóy River reaches the AC range, most gravel is contributed by the AC and SC ranges (Deaton and Burkart, 1984), implying that headwater-derived gravel has been trapped and/or comminuted before reaching the AC range. The Selegua River displays the steepest steepening as it crosses the AC range. Likewise, it reflects the fact that its headwaters are located in an area characterized by shallower slopes and underlain by a basement rocks, with make the river comparatively gravel-starved.

Such limited access to erosion tools are compensated by a larger increase in gradient (Sklar and Dietrich, 2006). The small drainage area of the Selegua River overall also makes it prone to detachment-limited behaviour (Brocard and Van der Beek, 2006).

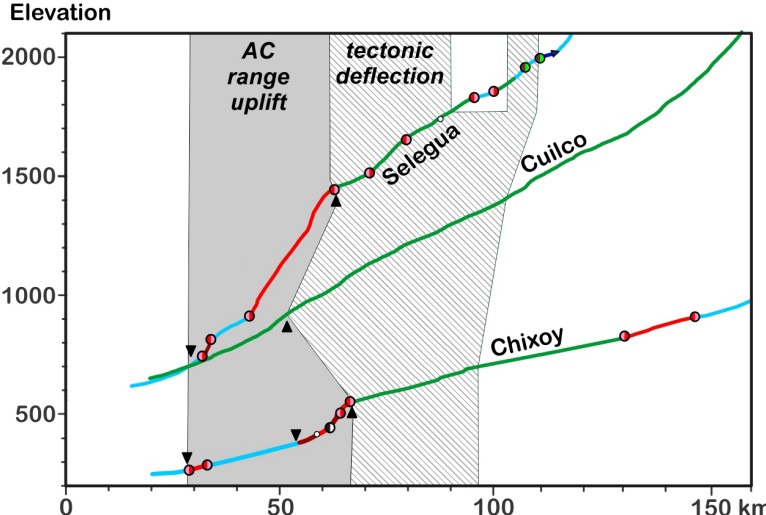

Figure 15. Long profiles of rivers transverse to the AC range: Chixóy, Selegua, and Cuilco, with indications of reaches affected by vertical rock uplift (grey area), and those affected by horizontal lengthening along the Polochic fault (hatched area).



### 5.2.2. Stalling of incision at the base of the SC range

765        Inception of uplift along the AC range modified the dynamics of the rivers that crossed the range, initiating accelerated incision across the AC range, and a slowing down of incision farther upstream  This was followed, along many of these rivers, by a phase of river aggradation before their abandonment, consecutive to the reorganization of their drainage (Brocard et al., 2011). In Plio-Quaternary times a similar evolution affected the rivers flowing across the northern side of the SM range. Normal faulting spurred valley infilling, upstream of the rising structures, followed by drainage reorganization

(Brocard et al., 2012). Aggradation upstream of a rising structure can be viewed as a transient, adaptive response of rivers to a changing uplift pattern along detachment-limited rivers (Brocard et al., 2012; van der Beek et al., 2002; Attal et al., 2008), or as an equilibrium response, under a stable uplift field, along transport-limited rivers (Humphrey and Konrad, 2000). The steepening of river profiles in response to accelerated rock uplift across the AC range generates a transient phase of surface uplift, upstream of the steepened reaches, during which incision does not counterbalances rock uplift. If the rock uplift rate

of the SC range does not increases at the pace of the uplift of the AC range, such uplift of river profiles requires a decrease in river incision rate upstream of the AC range. In effect, river incision slowed down considerably before drainage reorganization, but has remained extremely slow ever since (Brocard et al., 2011). This stalling of incision has allowed potentially up to 1,000 m of surface uplift, which is the current average elevation of valley floors, along the northern base of the SC range (Fig. 16a,b). The resumption of river incision, expected to mark the return to dynamic equilibrium, has not

occurred along the SC range. This implies either that river steepening is still under way across the AC range, and that dynamic equilibrium between river incision and rock uplift has not been reached yet, or that other processes prevent the resumption of erosion rates. Of two potential causes, the first one is climatic: it results from the aridification of the SC range. The second possible cause is tectonic, and results from the fact that the Cuilco, Selegua, and Chixóy River are deflected by the Polochic fault (Fig.15). Cumulative slip along the Polochic fault has generated a left-lateral deflection of these rivers,

lengthening of their courses by 25 km. The maintenance of a downstream gradient sufficiently steep to along at least the downstream dispersal of river bedload, if not incision, along the lengthened reaches, requires additional surface uplift, upstream of the tectonic deflections, which may contributes, in part or in whole, to the current passive surface uplift of the SC range.

       The similar stalling of incision along the Motagua valley over the past 7 My cannot be ascribed to the rise of a

tectonic structure across its course. It can result, too, from the establishment of aridity along the central part of the valley (Machorro, 2014), but it also may be driven by bottom-up processes. In this case, the floor of the Motagua valley may have risen by lengthening of the Motagua River course, driven by the progressive emergence, at its downstream termination, of transtensional basins in the Caribbean Sea (Carballo-Hernandez et al., 1988). The present-day difference in elevation (200-





600m) between the floor of the Motagua valley, south of the SC range, and the floor of the Chixóy basin, north of the SC

range, could be inherited from before 7 Ma ago, or could have been acquired, over the past 7 My, by 5° southward-tilting of the SC range.

**5.3. Effects of climate and tectonics on the celerity of migrating knickpoints**

Hillslope erosion is expected to increase downstream, across upstream-migrating knickpoints. A larger increase in hillslope erosion is expected to reflect faster upstream knickpoint migration (Brocard et al., 2016b). In the study area, the knickpoints that dissect the low-relief uplands do affect hillslope erosion rates, but this increase is much larger in the AC and SM ranges than in the SC range (Fig.10 and 13c). The lack of substantial increase in the SC range suggests that knickpoints are almost immobile there. Given that these knickpoints are not located near any obvious site where they could have

nucleated, it is likely that they formed some distance downstream. Alternately, they could have formed in situ, by some yet unidentified process. If these knickpoints formed farther downstream, they necessarily migrated faster in the past, before stalling at their current location. Knickpoint celerity is influenced by bottom-up processes, such as the rate of base level fall (Whittaker and Boulton, 2012). Base levels has been stable over the past 7 My along the northern flank of the SC range, and along the southern flanks of the SC and SM. By contrast, base levels have fallen rapidly along the northern flanks of the AC

and SM ranges, due to tectonic separation between the Petén lowlands/Central Depression of Chiapas and the AC range, and between the Izabal releasing bend and the SM range. Such differences may have driven, in part or in whole, the differences in celerity from a range to the other. Nonetheless, knickpoint celerity is also controlled by top-down processes such as the amount of water runoff and sediment discharge delivered from upslope (Crosby and Whipple, 2006; Brocard et al., 2016b; Whittaker and Boulton, 2012). The reduction in rainfall over the SC range, resulting from the >3,000 m surface rise of the

AC range, can therefore be expected to have also contributed to the slowing down of erosion waves along the SC range (Fig. 16a,b). No [10]Be measurements are available along the northern side of the SM range, but the abundance of precipitation implies that knickpoints are expected to migrate faster there. Most of the surface uplift the SM range occurred in concert with that of the SC range (as documented by the El Jute basalt). However the evolution of the northern flank of the SM range differs from that of the SC range in that it likely became wetter with time, in response to the channeling of moisture along

the Lake Izabal releasing bend. The releasing bend nucleated in the east, in middle Miocene time, steadily growing westward ever since (Bartole et al., 2019). Tectonic subsidence in the bend opened a topographic gap between the SM range and the mountains located farther to the north. Fluxes of Caribbean moisture are channeled along the gap, from which they rise along the northern flank of the SM range (Fig. 4, (Brocard et al., 2012)). The channeling of moisture along the northern side of range may have contributed to the growth of the rain shadow along its southern flank (Fig. 16c). The prominent knickpoints





of the northern flank are therefore likely migrating faster than their southern counterparts, under the influence of this relief-controlled precipitation regime.

The rise of the SM, SC, and AC ranges, and thereby for the spatial distribution of precipitation, has been driven by a tectonic displacements transverse to the strike of the plate boundary, during phases of enhanced transpression and transtension. The dominant, left-lateral component of the plate motion does not generate surface uplift. Still, it may alter the

distribution of precipitation by laterally displacing topographic obstructions over time. The Polochic fault moves the AC range westward relative to the SC range. By removing this topographic obstruction, the Polochic fault will ultimately re-expose the SC range to the Caribbean moisture. The velocity of the sliding (25 km in 7 My, (Authemayou et al., 2012)), however, is small compared to the length of range. Besides, the AC range may keep growing vertically and lengthwise such as to keep pace with its lateral offset. Left-lateral tectonics has therefore a limited impact in this specific case, on the

evolution of erosion rates by top-down processes.

Finally, it must be noticed that the aridification of the SC range also requires an efficient containment of the Pacific moisture by the Central American volcanic arc. The arc today supports stratovolcanoes more than 4,000 m high that rest on a crystalline Palaeozoic basement up to 3,500 m high. The arc currently efficiently contains the Pacific moisture. Its topographic evolution and uplift chronology are unfortunately fully unconstrained, and do not allow us to assess their

influence on the climate of central Guatemala over time.

## 5.4. Origin of the migrating knickpoints

Clusters of headward-migrating knickpoints are observed at mid-elevation along most of the studied mountain

flanks, downstream of the clusters of knickpoints that dissect the low-relief uplands (Fig.12). These mid-elevation knickpoints likely migrate at different celerities and have been generated by different processes.

Some scattered headward-migrating knickpoints dot the northern, southern, and northwest flanks of the AC range (Fig. 12b1,2). Along its northern flank, these knickpoints lie upstream of a broad area of stream steepening, over the hanging wall of the Ixcóy fault. They could mark the front of a wave of accelerate incision, triggered by an acceleration of differential

tectonic uplift (Whittaker et al., 2007) across the Ixcóy fault, while the overall steepening may reflect increasing uplift rate in the downstream direction, at dynamic equilibrium (Kirby and Whipple, 2001). The NW flanks of the AC range differs in that no tectonic separation of the Maya surface occurs at the base of the NW flank, where the Maya surface is affected by a kink fold (Fig.2 and 4). Mid-elevation migrating knickpoints there could have been produced by river steepening, without nucleating over a specific tectonic dislocation (Willenbring et al., 2013). Similarly, river steepening could account for the

two imbricated waves of erosion observed along the paleovalley of Quilén Novillo-Chancol, on the southern side of the range (Fig.8a).





Knickpoints are also observed at the base of the SC range, near the Chixóy River, at the downstream end of rivers that drain the N side of the range. These knickpoints, which are located downstream of the pediments, are restricted to the Chixóy Basin and are not observed along the northern flank of the SM range (Fig. 12b2). They could therefore result from the recent propagation of a wave of incision along the Chixóy River, which hosts a potentially migrating knickpoint slightly farther upstream (downstream of Sacapulas). The Chixóy River knickpoint, however, could also be lithogenic, or could result from uplift in the footwall of a normal fault near Sacapulas. This later fault is probably no longer active, as its scarp shows no indication of recent slip. The Chixóy River has established its range-parallel course between the AC and SC range ~7 My ago, during the reorganization of the drainage, as a collector at the lowest point in the saddle between the two ranges (Brocard et al., 2011). Today, however, the axis of this saddle is located 6-8 km farther south, suggesting that the saddle has migrated as a result of the enlargement of the AC range. The knickpoints located very near the Chixóy River could therefore have a tectonic origin, is a result of renewed uplift resulting from this enlargement, rather than spearheading a recent wave of incision.

A prominent cluster of headward-migrating knickpoints is located 500 m farther up the northern flank of the AC range (Fig. 8b and 12a2). If this wave of erosion was initiated by a distal tectonic event, they likely initiated over the first major tectonic dislocation encountered in the downstream direction, namely the Polochic fault. If so, the event that generated them would predate the uplift of the AC range. Indeed, only 20-90 m of incision has occurred over the past 7 My at the downstream end of these rivers (Brocard et al., 2011), and the knickpoints have typical amplitudes of 200-300m (Fig. S4-3). They could have been generated by accelerated differential uplift across the Polochic fault (Fig.16a), and their migration up the Chuacús range would have considerably slowed down over time, through a combination of decreasing upstream drainage area (Crosby and Whipple, 2006), and aridification of the SC range, and would subsist today, almost frozen, along the northern side of the SC range (Fig.16b). That the base of the knickzones lies in close association with the pediments that formed along the northern base of the SC (Fig. 8b and 12a2) further suggests that the knickpoints and pediments may be genetically related. If the process of upstream-directed growth of pediments (Pelletier, 2010; Strudley et al., 2006) proceeds faster than slope lowering farther upstream, they could generate the formation of knickpoints at the base of the range. The steepening of these knickpoints would occur by faster migration of their concave-up base (corresponding to the pediment apex) than their convex-up upper lip. Such evolution of headward-migrating knickpoint profiles is expected when river incision is controlled more by river discharge than river gradient (Weissel and Seidl, 1998; Tucker and Whipple, 2002).



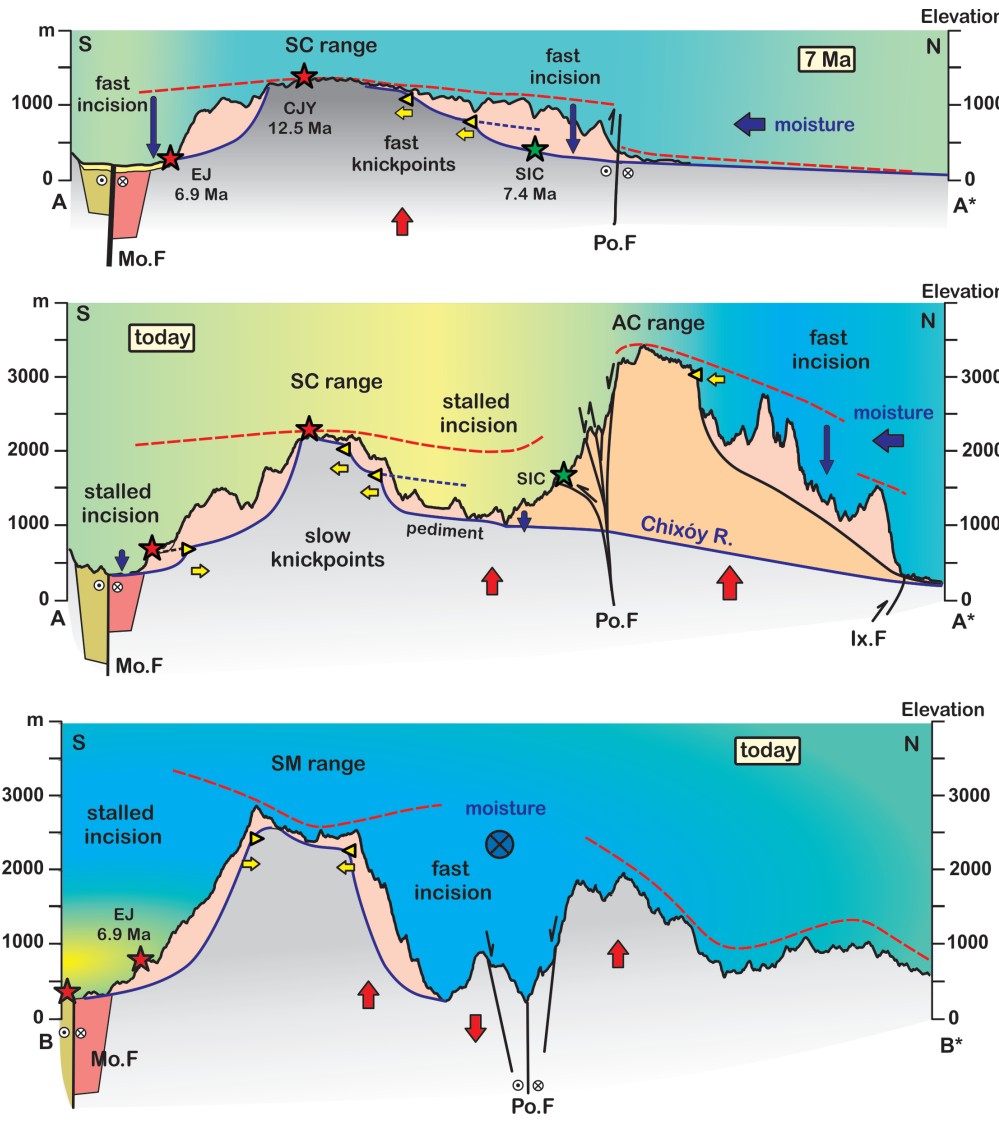

Figure 16. Development of the central ranges of Guatemala. Along profiles A-A* and B-B* (see Fig. 13a for location).

Mid-elevation headward-migrating knickpoints along the southern flank of the SM-SC range are mostly found

890 south of the SC range, along the upper reaches of the Motagua River. These migrating knickpoints lie at the head of valleys

incised into degraded low-relief pediments that are, at places, covered by ignimbrites (Fig. 12a1). These ignimbrites are not

dated, but considering that they hang above the Motagua River about the same elevation as the El Jute basalt, they could be

late Miocene in age, an age consistent with the age of other ignimbrites farther north (Brocard et al., 2011) and farther south

in Central America (Jordan et al., 2007). Downstream convergence of these ignimbrite with the Motagua River, and coeval





895 steepening of the Motagua River upstream of the convergence point suggest that renewed incision is taking place in the headwaters of the Motagua River. It is probably driven by uplift in the Central American Volcanic Arc, which forcing is difficult to track farther upstream, because the Motagua River course becomes increasingly chocked by the recurrent deposition of pyroclastic aprons, emplaced during caldera-forming eruptions along the volcanic arc (Wunderman and Rose, 1984). The last such eruption disrupted the course of the Motagua River 84 ka ago. Primary pumice flows deposited 100-200

900 m of pumice along the base of the SC range (Fig.12a), and several tens of meters of reworked pumice all along the base of the SM range, (Tobisch, 1986). This short-lived event did not generate knickpoints along the tributaries of the Motagua River, become most streams quickly incised the pumiceous deposits down to their pre-eruption level. The pumice, however, altered the upper course of the Motagua River, such that the river has epigenetically incised its upper reaches in harder volcanic substrates, generating knickpoints. At the other, downstream end of the Motagua River, sea level variations had no

905 noticeable impact on the studied reaches of the Motagua River, upstream of Los Amates (LA, Fig. 12a)., such that the Motagua River remains a stable base level for the streams that drain the southern flank of the SM-SC range.

 The northern flank of the SM range stands out for its absence of mid-elevation knickpoints. They may have never existed, or could have migrated faster upstream, as a result of the high precipitation along this flank of the range, merging with the high-elevation knickpoints that incise the ledge of the low-relief uplands. The stacking of erosion waves has been

910 proposed to account for the presence of some prominent knickpoints elsewhere (Crosby and Whipple, 2006).

### 5.5. Pediment formation

 A series of pediments has formed between the AC and the SC range, where incision rates have been the slowest

915 over the past 7 My (<30 m/My, Fig. 9), and where climate is the driest (Machorro, 2014). They have developed on granite and slate of the Santa Rosas Fm., as well as on schists and gneisses of the Chuacús Fm. They are currently extensively covered by the Los Chocoyos pumice, a volcanoclastic unit deposited during an eruption of the Atitlán caldera 84 ka-ago (Brocard and Morán, 2014; Rose et al., 1987). This recent depositional event gives the impression that the pedimented areas are sedimentary basins, overlain by thick sedimentary fills. Actually the pedimented surfaces are exposed by erosion at

920 shallow depth, below the pumice (e.g. Fig. 8b). Pediments tend to form in semi-arid landscapes, and require a stable base level (Pelletier, 2010). Along the northern side of the SC range, their apex are located immediately downstream of the detachment-to transport-limited transition, marked by concave-up knickpoints in the river profiles (Fig. S4-3, (Whipple and Tucker, 2002)), where lateral planation starts to dominates over vertical incision (Brocard and Van der Beek, 2006). In this context, the apex of these pediments can be seen as transient, upstream-migrating knickpoints marking a decrease in incision

925 rate (Baldwin, 2003).





Other pediments have formed in a similar setting, 200 km to the NW over the crystalline basement of the Sierra Madre de Chiapas (Authemayou et al., 2011a), in a dry area, isolated from the Pacific moisture by the Sierra Madre de Chiapas to the SE, and from Caribbean moisture by the Sierra de Chiapas in the NW. The Central Depression of Chiapas act as a stable base level for these pediments, and therefore appears to be passively uplifting, upstream of the Sierra de Chiapas, with respect to the Yucatán lowlands, much like the Chixóy Basin, upstream of the AC range, while the still uplift Sierra de Chiapas undergoes topographic decay along its NW flank.

The tectonic defeat and drying of ranges located on the leeside of newly uplifting ranges is expected to mark the nucleation and lateral growth of orogenic plateaus above crustal accretionary wedges (Sobel et al., 2003; Garcia-Castellanos, 2007). The decrease of topographic relief over the plateau interior combines a weakening of the crust, which does not allow the lithosphere to support tall reliefs, and the filling of intramontane basins with alluvium. We show here that pedimentation is an important contributing process, especially during the initial stages of topographic lowering, when the rivers draining the drier parts of the orogen are still connected with the foreland, allowing the dispersal of the sediments. In the case of fully-developing orogenic plateaus, increasing arid conditions during later stage eventually promote the disintegration of the river drainages, the disconnection of their base level from surrounding lowlands, and the full retention of erosional products within the intramontane basins.

## 6. Conclusions

- The radiometric $^{39}$Ar-$^{40}$Ar dating on volcanic rocks confirms earlier studies suggesting that the mountains of Central Guatemala located the closest to the plate boundary (Sierra de las Minas and Sierra de Chuacús) rose and were incised during the middle-late Miocene (12-7 My ago). Since then, the SM and SC ranges have stopped rising relative to the surrounding valleys (Motagua valley and Chixóy Basin), while the Altos de Cuchumatanes range, located farther north, has been uplifting and eroding.

- Strong uplift of the AC range over the past 7 My relative to the SC range results from oblique contraction combined to erosional unloading. The northern flank of the SM range has been gain some relief from the continued and westward propagation of the Lake Izabal releasing bend.

- Detrital terrestrial $^{10}$Be concentration in the bedload of rivers draining catchments distributed throughout these ranges show that current hillslope erosion rates mimic the current patterns of stream incision: they reach 300 m/My in the AC range, but are commonly lower than 100 m/My in the SC-SM ranges. In the AC range, hillslope erosion equals stream incision. In the SC range and along the southern flank of the SM range, hillslope erosion outpaces base level lowering, implying an overall trend of topographic decay.



- Current hillslope erosion rates coincide with the current distribution of precipitation. Precipitation is strongly controlled by topographic obstructions resulting from the rise of the SM and SA ranges, which intercept the Caribbean moisture. Precipitation is high along the northern flanks of the AC and SM ranges, while the southern flank of the SM range and both sides of the SC range lie within rain shadows. Fossil vegetation preserved in the Chixóy River basin documents a wetter climate 7 My ago, when the rain shadow was less pronounced or absent.

- In this context, the slow current hillslope erosion in the SC range appears to result from the rise of the SM and AC range and the development of these rain shadows. It can be hypothesized that hillslope erosion rates in the SC range were higher before the uplift of the AC and northern SM range started. The slowing of river incision at the base of the SC range and on the southern side of the SM range, because they occurred as the uplift of the AC range started, can be interpreted as the shutting down of erosion as a result of the growing rain shadow.

- The defeat of many rivers draining the SC range across the AC range, during the early stages of its growth, and the temporal coincidence between this defeat and the overall cessation of stream incision in the SC range further suggests that tectonic uplift along the AC range has switched off river incision upstream of the AC range, allowing the SC range to passively rise by up to 1,000 m, without undergoing significant erosion.

- The slowing of erosion over the SC range by top-down (precipitation), and by the bottom-up transmission of incision along streams crossing the AC range, has resulted in the slowing of upstream-migrating waves of dissection. Some of the headward-migrating knickpoints dotting the northern flank of the SC range may predate the rise of the AC range.

- Such shutting down of stream incision and hillslope erosion by top-down (precipitation) and bottom-up (tectonic steepening) processes provides a field illustration of mechanisms involved in the lateral growth of orogenic plateaus.

**Author contributions**

Gilles Brocard: project design, river [10]Be sampling (Cuchumatanes, Sierra de las Minas), [10]Be sample processing (Cuchumatanes, Sierra de las Minas), river segment analysis. Manuscript preparation, with contributions from all authors. Jane Willenbring: [10]Be sample processing (Sierra de las Minas, Sierra de Chuacús). Tristan Salles: river profile segmentation. Mickael Cosca: [40]Ar/[39]Ar dating. Axel Guttierez and Noe Chiquín: [10]Be stream sampling (Sierra de las Minas, Sierra de Chuacús). Sergio Morán-Ical: field work coordination. Christian Teyssier: project design and coordination.

The authors declare that they have no conflict of interest.

**Aknowledgments**



This work was supported by the Department of Geology and Geophysics at the University of Minnesota,
Minneapolis, UMN grant-in-aid 1003-524-5983, and by the Swiss National Science Foundation grant 200020-120117/1.

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
