# Peer review of "Tectonically- and climatically-driven mountain-hopping erosion in Central Guatemala from detrital 10Be and river profile analysis"

_Earth Surface Dynamics, 2020_

## Referee Comment (RC1) · Anonymous Referee #1 · 16 Dec 2020

General comments

The manuscript addresses landscape evolution in two parallel ranges in central Guatemala that were uplifted 7 Myr apart in the Miocene. The authors use Ar-Ar dating to date volcanic deposits that cap parts of the range and to calculate incision rates, 10Be-derived catchment-averaged erosion rates and hilltop exposure rates, stream profile analysis and knickpoint identification, and topographic analysis. This is an impressive ensemble of methods, all of which are standard tectonic geomorphology tools that appear to have been applied correctly, and a large amount of new data is presented. The authors conclude that uplift of the second range created a rain shadow

that slowed erosion rates on the northern flank of the first range and stalled knickpoints. Unfortunately, the manuscript has two major problems that need revising before publication. The first problem is that it does not adequately synthesize the large quantity of data that is presented. Many of the figures are confusing and contain too much information, and too much detail is included in the text. I recommend omitting (or moving to the supplement or another paper) all information that is not crucial to testing one or two hypotheses. The second problem is that the main conclusion that the observed spatial variation in erosion rates is controlled by climate is not supported by the evidence. The correlation between the erosion index (which take into account precipitation) and erosion rates appears to be weak (no measure of correlation is presented) and drainage basin dynamics (which appear to be important here) are not considered.

In a chi map for the region (see Giachetta and Willett, 2018, A global dataset of river network geometry), one can see that the slowly eroding northern flank of the older range are tributaries to a large basin that is predicted to be losing drainage area (see bottom left of attached screenshot in which one basin has higher chi – warmer colors – than the surrounding basins). Higher chi means higher predicted steady-state elevation. Thus, the basin will need to steepen for erosion rates to keep up with uplift rates. In response, erosion rates slow and the region is passively uplifted. The geometric disequilibrium could be a result of initial geometry or potentially a result of lengthening of the mainstem along the strike-slip faults. Deformation of channels along the faults is mentioned, but how that lengthening might impact erosion rates upstream is not addressed. Furthermore, the mechanism by which precipitation would impact erosion rates is not clearly explained. Theory says landscapes adjust their erosion rates to keep pace with uplift, so spatial variations in precipitation (with uniform uplift) should manifest as differences in normalized channel steepness (ksn), with steeper rivers in drier places, rather than differences in erosion rates. However, it's possible aridification resulted in a dramatic decrease in K (erosional efficiency of a river) which would lead to an increase in response times in rivers draining the drier zone, and so you wouldn't see steeper rivers in drier zones because the basin has not had time to respond (and

yes you would also see stalled knickpoints as they do). Erosion rates would drop below uplift rates in the upper reaches that have lost precipitation but not steepened yet, and adjacent basins that have not lost precipitation would start to capture area off the basins that experienced aridification. If I were testing the hypothesis that aridification resulted in a slowing of erosion rates, I would also ask the questions: 1) Can the spatial variations in erosion rates be explained by drainage basin dynamics alone? 2) Is the whole region that experienced aridification now experiencing area loss?

Specific comments

The section on knickpoint analysis could be made more concise. I don't think you use any knickpoints that aren't migrating knickpoints, so you can just say you eliminated other knickpoints using image analysis. I think you can get rid of the section explaining all the different types of knickpoints. Are there knickpoints in multiple basins that you can trace to a single origin? If so you could use this as evidence for a change in K in basins that experienced aridification. Do the knickpoints you claim are stalled in SC cluster at the same chi value (and hence, have a single downstream origin)?

How were erosion rates for nested basins dealt with? Was a mixing model used to calculate the erosion rates from the additional area added with each larger area? If not, I think it would be a good idea to do this. Also, clarify why nested basins were chosen and what hypotheses they were used to test.

There is a lot of background and discussion mixed in with the results and it makes it hard to discern exactly what was done as part of this study. Keep results separate and move interpretation of results to the discussion.

3.3 – A "hillslope erosion index" is presented but then stream discharge and stream gradient are used, suggesting channel networks not hillslopes are actually being analyzed. Clarify what you are measuring and why.

Throughout: Ma should be used for "millions of years ago" and Myr should be used for

millions of years. See Aubry et al (2009) Terminology of geological time: Establishment of a community standard

Line comments

169 fig 4 not fig 3 398 and with uniform uplift, otherwise K matters 654 I do not see a correlation, nor are correlation measurements presented for either b or c 711 theory is off here, see general comments 774-776 how does uplift lead to a slowing of incision rates upstream? 801 Not intuitive. Why doesn't it reflect the magnitude in change of forcing? Clarify. 807-808 I'm not familiar with Whittaker and Boulton, 2012, but stream power theory states celerity only depends on uplift rates if n is not unity (Whipple, 2001) 853 what? Knickpoints are areas where the rivers are anomalously steep so how can you say they are caused by river steepening? 873 what does it mean for a knickpoint to have an amplitude? 937-940 citation?

Figure comments

Fig 1 DEM resolution not great, also figure shows elevation not relief so title is confusing. Fig 2/3 recommend using the same colors for the paleosurface throughout Fig 5 make legend transparent also so colors match Fig 8 I find this figure very overwhelming. If you keep it, the legend needs reworking or it at least needs to be more clear that the numbers are knickpoints. Leaving off lithology would help. Fig 9 Do you say how many incision measurements you have? If so I missed it. Could make the boxes proper box plots rather than just showing range. Fig 10 This figure is confusing. Would be more helpful to see rates on a map. Fig 11 Clarify what this figure shows. Fig 13 How were the 10Be rates normalized? Recommend one map showing erosion rates and put the map of the erosion index in the supplement. Add main divides.
* * *
[Figure]

**Fig. 1.** Chi map of region from Global Chi dataset

---

## Referee Comment (RC2) · Paul J. Umhoefer (Referee) · 17 Dec 2020

General: This manuscript addresses the history of landscape and river evolution in the mountain ranges across Gautemala and southern Mexico where the North America – Caribbean transform plate boundary meets the Cocos – Central American convergent (volcanic arc) boundary. The area is tectonically complex at present and in the past because of this setting. More relevant here is that the faults are large and have grown over millions of years; therefore, the topography has changed because of the faulting and the resultant climate and landscape processes have reacted to this evolving setting in complex ways. The history studied here stretches over 12 Myr from the middle

Miocene to the modern, and therefore a multitude of methods are used across geomorphology and tectonics. I find this a fascinating and potentially enlightening region, but its complexity challenges the researchers when writing papers for an outside audience. The main point of the paper is the variable response of erosion and rivers on two adjacent ranges that were uplifted sequentially in the past 12 Myr across the study area. The authors argue for two main conclusions – that the younger range erodes faster, and erosion correlates with the amount of precipitation, whose distribution is controlled by the rising ranges themselves. They present numerous alternatives to these main conclusions, but in a manner that seriously distracts from the presentation of the meaning of the results. There is a good paper hidden in this manuscript, but the present version needs major revisions. The writing is fine in the Introduction, Methods, and Results sections, though these sections need one more thorough editing. But the writing is much more opaque in the Discussion, and that section needs some serious rewriting or reorganization. The Discussion section badly needs clear topic sentences leading the paragraphs. Many sections of the Discussion include multiple conclusions but not a clear statement at the beginning or end on which was the dominant process, or conclusion. Here are my suggestions for improving the Discussion. First, part of the problem may be the approach to the conclusions of this study using top down and bottom up controls - at first this seems logical, but it also seems to unnecessarily make the author's discussion of the results and the conclusions more complex than necessary. There is a story to tell here that has more universal implications for landscape evolution processes elsewhere, but first we need to hear the local story told in a succinct, coherent fashion. Second, the relation of the main conclusions to the study results in the Discussion section is not at all clear. The authors start the manuscript in the Abstract with two main conclusions from the consecutive uplift of two adjacent ranges since the mid Miocene. The Discussion has no focus on these larger conclusions, but instead wanders across many topics related to the study area. I would completely rewrite the Discussion to start with a section that focuses like a laser on which results from this present study support your main conclusions. Then it would be fine to add a section in

which you can digress to a discussion of alternative conclusions, but I would keep this section short. I would conclude the Discussion with a section on a succinct summary of the 12 Myr history of the landscape evolution of the study area that makes clear where the new results contributed. Many of the figures need improvements – much better explanations (legends) elucidating the details in the figures; more complete captions; better links between figure content and the text. See my comment on figure 5 as an example. And the manuscript needs more citations of the figures in the text. One more regional figure is required for the Introduction and setting sections, and much of the discussion of tectonic controls. There are many references to the larger faults and river drainage areas that lie outside the many map figures of the study area. Most readers will require a new intermediate scale map that covers the region around the study area. That new map should show the larger context of the topography and faulting - I think a topo base (geomapapp?) with the main faults is sufficient. The rough area of such a new map should show the offshore beyond the trench to the west and well into the Caribbean to the east - that is a good area of S Mexico and Honduras.

Specific Comments: (I did not edit the manuscript in detail – the manuscript needs one thorough edit) A few comments relevant throughout the manuscript: • Check for use of modern usage of unit abbreviations throughout: millions of years ago = Ma; million years duration = Myr, so rates are m/Myr. Ages are XX Ma. • Watch for overuse (or any use?) of anthropomorphic language when referring to geology and geomorphologic features. I give a few examples below. • I give only a few examples here, but there are many, many examples of where the writing could be shortened with no loss of content or meaning.

Abstract: It would be much better to get the reader into the link between mountain range uplift and other parameters from the start by adding the names of the ranges in the Abstract: "two parallel, closely spaced mountain ranges formed during two consecutive pulses of single-stepped uplift, one from 12 to 7 Ma (Sierra de Chuacús), and the second one since 7 Ma (Altos de Cuchumatanes)."

Figure 2 and 3: more clearly label the Maya surface. Lines 89-91: The manuscript is longer than it needs to be. One example is writing that could be more concise with one example provided: In Middle Miocene time, the topography of Central Guatemala was very subdued . Remnants of this low relief (referred to as the Maya surface) still cap many mountaintops across the study area (Fig.2). The low middle Miocene relief formed from the topographic due to the decay of Eocene folds (Authemayou et al., 2011b; Brocard et al., 2011). Lines 72-73: Delete this sentence – you just said this in the line 51 paragraph or tell us for what purpose did you investigate these ranges? line 113: "westward decrease in the length of river deflections along the Polochic fault" this is not clear – if anything the black arrows in figure 2 are increasing in length to the west. Do you mean "increase"? If not, this interpretation needs to be better explained. line 114: "consecutive to an earlier rise" clumsy use of consecutive – better(?)  to say "caused by an earlier rise" or "following an earlier rise" Line 121: Ixcán fault is on figure 5 and not on figure 4. Line 131-135: Cite figure 1 for the lake. And again a map that covers a larger area would show the releasing bend. And the eastern end of the Motagua fault is not on any map. Line 138: "Slip on the Motagua fault is purely left-lateral today..." Despite the major bend through the study area? Are you sure it is not transpressional along its western bend? Line 140 paragraph: Cite figure 5 and add the name Subinal Fm to figure 5. Line 144: high angle faults instead of steep angle faults Line 157: cite your figures more – here cite figure 1. Line 159: up to the editors, but most journals would prefer non-anthropomorphic words to replace "benefits" and "at the expense of" a catchment. There are many examples of this writing style across the manuscript. Line 169: should be figure 4, not 3. Line 195 paragraph and Figure 5: Add the formation names to the figure 5 explanation. Add the metamorphic to N America basement line on figure 5. Line 207: "processes in its carbonate rocks generates complex..." make it clear: "processes in the carbonate rocks of the AC range generates complex..." Line 209: better: "over the carbonates in Late Cretaceous (Campanian) time." Many (most?) geoscientists don't know the time scale at the stage level. Line 243: do you mean the timing or rate or what parameter

**ESurfD**
of hillslope erosion? Figure 8: The four maps are A, B, C, D (capitalized). Lithologies would be more clearly labeled as a, b, c, d rather than Greek letters.

Section 3.6: most of the knickpoint classification would be better moved to supplemental material. Then you could just keep the text on the other methods related to the knickpoint analysis.

Line 446-447: cite your figures 2 and 3 for Maya surface. Line 452-453: I would add the new ages to figure 2 and cite that here. Does the incision have to be immediately after the 12 Ma lahar? How much after can it be – that is what is the constraint on the younger side of the window of possible incision timing? Where is the Cuilco River valley? The Colotenango valley? Line 460: here and elsewhere: mixing abbreviations SC range etc with the occasional Sierra de Chuacús spelled out is more confusing than sticking to one nomenclature. Line 470: an elongate basin is a variation on a trough – drop the trough and just refer to it as a basin. There is no "trough basin" in the world of sedimentary basins. Lines 474-483: much of this argument is not convincing, or at least a role (major?) for faulting cannot be disregarded based on your points here. Sediments bypassing an actively faulted basin are common with high enough sediment flux and an overfilled basin. The climate – erosion machine can easily outpace subsidence from faulting. You may well be missing evidence for normal faults cutting alluvial fans because that evidence lies deeper in the subsurface of the basin – this happens all the time in young basins where only the upper alluvial fans are exposed. The last point is valid but with a very large offset strike-slip fault, how much differential offset across it would one expect? Might the two paired strike-slip faults work in unison to result in roughly the same elevation of old surfaces across the Motagua fault? Line 490: "Using the modern gradient of the Huijo River valley as a proxy...". But above in the text you give results showing that at 6 Ma the SM range incision was slowing greatly over a few Myr. So is it valid to compare the modern gradient to one at 6 Ma? Was the river gradient also higher at about 6 Ma? Section 4.1: Overall I find the conclusions in this section solid.

**ESurfD**

Interactive
comment

Discussion: Line 645 – 647: Would a simple graph of study-derived erosion rates vs precipitation at that site be effective here? Section 5.2.1.2: This section is emblematic of the mixed conclusions resulting partly (largely?) by the organization of the Discussion. But this section (and others) also needs a clearer initial statement of the conclusions of your study relative to the topic of the section. You state up front in this section that "we conclude that the increase observed across the AC is a response to faster tectonic uplift" – Im assuming the increase you refer to is the increased river gradient across the AC range. Then you spend most of the section discussing what seems like a better explanation of the control of the bedload type on the rivers. Figure 15: which parts of these three rivers do these profiles represent? What is point 0 km on the diagram mean? What are the symbols on the three profile lines? Section 5.3: this section is the epitome of the problem I have with the Discussion section as a whole. The section does not focus the reader on the results from the present study, and the conclusions from those results. Instead, the section is a jumble of alternative conclusions about the migration of knickpoints. In addition, to even consider the statements on the tectonics of the study area requires the new figure I asked for of a larger area.

Conclusions: I would repeat the main conclusions from the Abstract here, then organize the conclusions within that context. Line 975: The relation of the conclusions of this study to orogenic plateau growth is a stretch here because the authors do not have the space to make the case. I would drop this theme and this conclusion here and earlier in the paper.

---

## Author Comment (AC1) · 2 Feb 2021

Reply to interactive comments on Âń Top-down and bottom-up controls on mountain-hopping erosion : insights from detrital 10Be and river profile analysis in Central Guatemala

Overview

We modified the title such that it better reflects the contents of the paper. The new title makes more explicit reference to the contribution of climate and tectonics, and to the discussion on landscape evolution.

[Figure]

Based on remarks of referee #2, we rewrote the end of the introduction to clarify the structure and intentions of the paper, simplified the description of the study area (section 2). We reworked the structure of the discussion, expressing more clearly the respective contributions of precipitation (5.1) and river incision dynamics (5.2) to the slowing down of erosion over the old range. Following remarks from referee #1, we provide a more detailed assessment of the contribution of drainage dynamics to the evolution of incision rates in section (5.2). We then continue the discussion (5.3), like in the first version, with a review of some of the consequences of the slowing down of erosion rates on the topographic evolution of the old range. However, we sieved out interpretations that are mostly of regional significance, moving them to Supplement 4. We believe that the paper is much tighter this way.

Replies to anonymous referee #1

We thank anonymous referee #1 for his comments, which helped us reorganize the paper and make the effects of climate and tectonics more clearly presented.

Replies to the general comments:

Overview Anonymous referee #1 shows a great fondness for drainage divide migration at drainage dynamics at equilibrium, and holds strong views against precipitation control on erosion. In short, the referee seems to ascribe the slowing down of erosion along the northern flank of the older range to "drainage dynamics", or, in mechanistically more explicit terms, to the transient response of the drainage to the enlargement of the range, presumably from one state of dynamic equilibrium (uplift of the old range) to another (uplift of both ranges). The referee therefore dismisses the "top-down", climate driven control on erosion, claiming that only drainage reorganization is at work. From the data at hand, we find that both processes are at work, and state this more clearly in the discussion. The paper is not focused on quantifying their respective merits, but rather on how they team-up to produce the range-hopping erosion pattern.

Referee #1 claims that we ignore the effect of drainage dynamics even if we actually

devote an entire section of the paper to bottom-up processes (specifically the effects of the rise of the new range, and of strike-slip tectonics, on the slowing down of erosion rates farther upstream). Referee #1 repeatedly analyses this non-equilibrium landscape within the framework and with the rhetoric of dynamic equilibrium. This has limitations when analysing such a complex landscape, which starts, and remains, far from equilibrium. As a result, inferences made at equilibrium are often disputable. The criticism of the effects of climate on hillslope erosion is based on objections regarding the meaningfulness of field data. Admittedly, field data are rarely optimal (here, due to issues of data acquisition in perilous circumstances). We introduced a test of significant that shows that these data do support a positive relationship between erosion rate and precipitation rather than otherwise. For some reason, referee #1 seeks to find support for drainage divide migration and drainage dynamics using a chi-map. The surprising use of the chi-map itself is detailed hereafter. Notwithstanding, referee #1, while questioning the increase in erosion from the old range to the new range, ignores the fact that the very same dataset clearly shows no significant difference in erosion rate on either side of the older range that would indicate drainage divide migration. Referee #1 likewise does not seem to bother that such migration would have destroyed the remnants of the old Miocene surface. The various pulses of uplift and incision that have affected this landscape have not transmitted their signal as far as the divide yet. In our earlier papers, we reconstructed in detail the evolution of the drainage, based on geomorphology and sediment provenance. It is beyond the scope of this paper but it also demonstrates the absence of the divide.

Specific replies to comments

The first problem is that it does not adequately synthesize the large quantity of data that is presented. Many of the figures are confusing and contain too much information, and too much detail is included in the text. I recommend omitting (or moving to the supplement or another paper) all information that is not crucial to testing one or two hypotheses.

Were are not testing one hypothesis against the other. We show how our broad array of field data is consistent with the influence of both processes, and what their morphological expression is. Simplifications have been introduced, however, which details are provided in the response to referee #2.

The second problem is that the main conclusion that the observed spatial variation in erosion rates is controlled by climate is not supported by the evidence. The main conclusion does not state this anywhere. It actually states that both climate and drainage dynamics are important (hence the title, which stresses the importance of both processes).

The correlation between the erosion index (which take into account precipitation) and erosion rates appears to be weak (no measure of correlation is presented)...

There is a threefold increase in erosion rates from the older range to the younger range, associated to a twofold increase in erosion index values. These data do not show, therefore, an absence of increase, nor a decrease, from the older to the younger range. In detail, the correlation is strong among the sites in the younger range, and weak among sites within the older range. In the discussion, we briefly discuss the absence of correlation within the older range (lines 667-671). We would happily discuss this in length, but it would distract the reader from the main plot.

... and drainage basin dynamics (which appear to be important here) are not considered.

In the very introduction (line 56-59), we stress that drainage dynamics is the primary motivation of this work, following our previous studies of drainage dynamics in the area (Brocard et al., 2011; 2012).

In a chi map for the region (see Giachetta and Willett, 2018, A global dataset of river network geometry), one can see that the slowly eroding northern flank of the older range are tributaries to a large basin that is predicted to be losing drainage area (see

bottom left of attached screenshot in which one basin has higher chi – warmer colors – than the surrounding basins).

It does not seem to bother you the least that the calculation of chi starts far away from the studied range, and therefore that the streams reach the range (in the upstream direction), with chi values that already "predict" drainage migration. The very use of this chi map is problematic anyway here, given the lack of spatial and temporal homogeneity and the lack of accurate weighting, on this chi-map, of precipitation, rock uplift rates, rock erodibility, fluvial processes, and base level stability effects. Considering that no of these parameters is spatially nor temporally constant, teasing out their respective contributions to the chi values is difficult. In the case at hand, chi values capture changes in basin shape and river length of streams that drain karstic lowlands (in the north), bypass lowlands, and feed subaerial sediment depocenters. If we could quickly rewind the evolution of chi-values at the divide, we would see that they are affected by high-frequency differential oscillations triggered by changes in sea level, an unequal increase in chi values on both sides of the divide, related to the lengthening, over at least the past 5 Ma, of the outlet rivers over their shelves in the Caribbean Sea and in the Gulf of Mexico, and then, within the range, a drop in chi-value on the northern side 7 My ago, resulting from the integration of many parallel rivers with elongated catchments, into a single, larger, more dendritic drainage, and then, likely, a drainage reversal of Rio Motagua, on the southern side, with a big increase in chi values. Therefore, the current differences in chi values across the divide, 1st do not mean much in terms of drainage dynamics, and, 2nd, in effect, never led to divide migration, as documented by the field data (clast provenance studies on the northern and southern sides, preservation of the Miocene paleo-surface along the divide).

Higher chi means higher predicted steady-state elevation. Thus, the basin will need to steepen for erosion rates to keep up with uplift rates. In response, erosion rates slow and the region is passively uplifted. The geometric disequilibrium could be a result of initial geometry or potentially a result of lengthening of the mainstem along

the strike-slip faults. Deformation of channels along the faults is mentioned, but how that lengthening might impact erosion rates upstream is not addressed.

These two points are actually addressed in section 5.2.2, and correspond to the "bottom-up" controls on incision in the older range. To follow your reasoning in the context of dynamic equilibrium, we need to view the rise of the new range as an enlargement of an equilibrium prism. In the new section 5.2.. we discuss is more detail how the rise of the new range results in surface uplift of the old range. Likewise, the impact of lengthening is addressed in the same section: "maintenance of a downstream gradient sufficiently steep (. . .) along the lengthened reaches, to convey the sediment load, requires additional surface uplift, upstream of the tectonic deflections, which may contribute, in part or in whole, to the current passive surface uplift of the SC range". In the new version, we provide an estimate of how much uplift can be ascribed to uplift and lengthening.

Furthermore, the mechanism by which precipitation would impact erosion rates is not clearly explained. They are clearly specified both when presenting the erosion index (methods), and then in the discussion of climate influences. Besides, the positive correlation between precipitation and erosion is commonly observed in temperate and subtropical climates.

Theory says landscapes adjust their erosion rates to keep pace with uplift, so spatial variations in precipitation (with uniform uplift) should manifest as differences in normalized channel steepness (ksn), with steeper rivers in drier places, rather than differences in erosion rates.

Precisely: Ksn values are high in the younger range, despite wetter conditions and higher erodibility. This supports the point that the range is uplifting faster (and, measurably, erodes faster). However, because rivers are not all detachment-limited, we considered that the use of the behaviour of Ksn could be misleading. This is why we analysed separately the behaviour of Ksn according to segments that are boulderarmoured or transport-limited.

However, it's possible aridification resulted in a dramatic decrease in K (erosional efficiency of a river) which would lead to an increase in response times in rivers draining the drier zone, and so you wouldn't see steeper rivers in drier zones because the basin has not had time to respond (and yes you would also see stalled knickpoints as they do). Erosion rates would drop below uplift rates in the upper reaches that have lost precipitation but not steepened yet, and adjacent basins that have not lost precipitation would start to capture area off the basins that experienced aridification.

. . . this is basically what we observe, including the start of a tendency for wetter catchments to capture dry catchments (the area of the Quaternary captures). We were very tempted to include a paragraph on the future evolution of the drainage, precipitations, and erosion, based on the current dynamics, but this would make the paper unnecessarily longer.

If I were testing the hypothesis that aridification resulted in a slowing of erosion rates, I would also ask the questions: 1) Can the spatial variations in erosion rates be explained by drainage basin dynamics alone? 2) Is the whole region that experienced aridification now experiencing area loss?

1: considering that both processes act in concert in reducing erosion rates, assessing their contribution may at best highlight the importance of one over the other, but certainly not disprove the contribution of one of them. Besides, higher precipitations would easily lead to higher erosion rates, and a faster topographic decay of the older range, despite low river incision rates due to drainage dynamics. Besides, incision and hillslope erosion have also stalled also along the southern, dry side of the old range, were no change in drainage dynamics is observed. We expose these facts more clearly in the discussion, after presenting evidence for both climate and tectonic controls.

2: drainage migration from the wetter to the drier is observed between the Chixóy (dry) and Panima (wet) basins. Still, some tectonic forcing contributes to this dynamics (Brocard et al., 2012). On the main range drainage divide, migration is not as simplistic as a SPL-equilibrium-blitz response predicts: drainage dynamics and climate did not trigger any substantial migration of the drainage divide. The presence of knickpoints, dissecting the middle Miocene surface, clearly shows that the pulses of incision generated over the past 12 My have not yet been felt at the divide.

Replies to the specific comments:

The section on knickpoint analysis could be made more concise. I don't think you use any knickpoints that aren't migrating knickpoints, so you can just say you eliminated other knickpoints using image analysis. I think you can get rid of the section explaining all the different types of knickpoints. Are there knickpoints in multiple basins that you can trace to a single origin? If so you could use this as evidence for a change in K in basins that experienced aridification. Do the knickpoints you claim are stalled in SC cluster at the same chi value (and hence, have a single downstream origin)?

The use of such approach is already presented when explaining how the selection was done. (considering that only 14 % of the knickpoints are most likely migrating, it is important to explain how the screening was done). As explained in the text, the ranges here are too complex to track knickpoints to a single origin. We have been able to do this more reliably elsewhere, for example, on in Puerto Rico (clustering of elevations), but even there, simple variations in bedrock erodibility generate a broad scatter in chi-values among knickpoints tied to a single origin.

How were erosion rates for nested basins dealt with? Was a mixing model used to calculate the erosion rates from the additional area added with each larger area? If not, I think it would be a good idea to do this. Also, clarify why nested basins were chosen and what hypotheses they were used to test. Mixing models are cool but are overkill here. They are not needed to show that the older range erodes more slowly than the new one.

3.3 – A "hillslope erosion index" is presented but then stream discharge and stream

gradient are used, suggesting channel networks not hillslopes are actually being analysed. Correct. We corrected stream to slope. The EI is meant there to capture overland flow on hillslopes.

Throughout: Ma should be used for "millions of years ago" and Myr should be used for millions of years. See Aubry et al (2009) Terminology of geological time: Establishment of a community standard We are aware of this rule, which varies from a journal to the next. We will follow ESurf's standard.

Line comments 169 fig 4 not fig 3 . Modified. 398 and with uniform uplift, otherwise K matters . Uniform uplift added 654 I do not see a correlation, nor are correlation measurements presented for either b or c. We added a correlation coefficient. 711 theory is off here, see general comments. You assume that the range gets dryer whereas it gets wetter there. 774-776 how does uplift lead to a slowing of incision rates upstream? As per the very theory of dynamic equilibrium and drainage dynamics you defend so much.

801 Not intuitive. Why doesn't it reflect the magnitude in change of forcing? Clarify. This is explained in the mentioned references.

807-808 I'm not familiar with Whittaker and Boulton, 2012, but stream power theory states celerity only depends on uplift rates if n is not unity (Whipple, 2001). . . . Only in this case, yes. It's not because you love the SPL that the real word is offered that only possibility.

853 what? Knickpoints are areas where the rivers are anomalously steep so how can you say they are caused by river steepening? We cite the proper reference here, see the modelling results of Nicole Gasparini.

873 what does it mean for a knickpoint to have an amplitude? Knickpoints are often described as waves of erosion, with a celerity. We replaced it by height

937-940 citation? They are the same as two lines before, we repeated them for clarity.

Figure comments Fig 1 DEM resolution not great, also figure shows elevation not relief so title is confusing. This DEM resolution was chosen because on higher resolution DEM, the shading prevents seeing overlays clearly. Fig 2/3 recommend using the same colors for the paleosurface throughout Color adjusted Fig 5 make legend transparent also so colors match We improved the color match.

Fig 8 I find this figure very overwhelming. If you keep it, the legend needs reworking or it at least needs to be more clear that the numbers are knickpoints. Leaving off lithology would help. We transferred text from the caption to the legend to make it easier to read.

Fig 9 Do you say how many incision measurements you have? If so I missed it. Could make the boxes proper box plots rather than just showing range. These are not box plots.

Fig 10 This figure is confusing. Would be more helpful to see rates on a map. Fig 11 Clarify what this figure shows. Fig. 10 is a neutral figure in the result sections. It provides rates values that could not be read on a map. Fig11: We clarified the caption.

Fig 13 How were the 10Be rates normalized? Recommend one map showing erosion rates and put the map of the erosion index in the supplement. Add main divides. The 10Be rates are not normalized. We understand you don't like EI, but the map nicely illustrates the predicted distribution of erosion rates through the study area, at the same scale as that of the introductory figures, which is informative for people interested in the area. This map is also a support to the cross sections of figure 16.

Replies to anonymous referee #2

We thank anonymous referee #2 for a careful review of the paper and we expose here which changes were implemented

Replies to the general comments:

The authors present numerous alternatives to the main conclusions, but in a manner that seriously distracts from the presentation of the meaning of the results. The writing

**ESurfD**
is much more opaque in the Discussion, and that section needs some serious rewriting or reorganization. The Discussion section badly needs clear topic sentences leading the paragraphs. Many sections of the Discussion include multiple conclusions but not a clear statement at the beginning or end on which was the dominant process, or conclusion. I would completely rewrite the Discussion to start with a section that focuses like a laser on which results from this present study support your main conclusions. Then it would be fine to add a section in which you can digress to a discussion of alternative conclusions, but I would keep this section short. I would conclude the Discussion with a section on a succinct summary of the 12 Myr history of the landscape evolution of the study area that makes clear where the new results contributed. I would conclude the Discussion with a section on a succinct summary of the 12 Myr history of the landscape evolution of the study area that makes clear where the new results contributed.

We simplified the presentation of the study area (section 2), transferring some of its material to the discussion section, and also simplified the presentation of methods (section 3) and results presentation (section 4). We reorganized the discussion (section 5), into three subsections, one presenting links between climate and erosion (5.1), the second one the effects of drainage dynamics (5.2), and then a section that discusses their combined effects on the current topographic evolution of the old range that are of most general significance (5.3: slowing down of headward-migrating knickpoints, pediment formation). We end up with a subsection and the implications for the drying down of mountain ranges and plateau formation.

One more regional figure is required for the Introduction and setting sections, and much of the discussion of tectonic controls. There are many references to the larger faults and river drainage areas that lie outside the many map figures of the study area. Most readers will require a new intermediate scale map that covers the region around the study area. The rough area of such a new map should show the offshore beyond the trench to the west and well into the Caribbean to the east - that is a good area of S Mexico and Honduras.

We added a map that clarifies the regional context, although we feel that it is more of regional significance here. Indeed, we have tried to avoid as much as possible providing information that distracts the reader from the two-ranges story, which is the focus of this paper. We have presented the intricacies of the regional, tectonic context in other papers.

Specific comments We used the specific comments to modify the text, through the thorough rewriting of various sections.

---

## Author Response (AR2)

**Replies to the associate editor's comments on manuscript 2020-80-R1**

We have implemented the edits suggested by the editor throughout the manuscript, and implemented stylistic simplification, and disambiguation throughout. These changes do not alter in any way the meaning of the sentences.

Please not that further changes may need to be implemented based on the internal review of the USGS (mandatory for the publication of the Ar-Ar ages).

More substantial changes, based on the editor recommendations, are explained hereafter

*1. Figure 2 : volcanic derangement*. It is actually explained at the end of section "2.2. Drainage evolution since the Middle Miocene", but we make more specific reference to figure 2 in this section when mentioning it

*2. Calculation of Chi without precipitation weighting*.

We now explain that the linearization was only aimed at locating the knickpoints.
It turns out that the choice of the concavity values had no effect on their location and number, and the profiles were well linearized, so we did not resort to more complex forms. Besides, the current pattern of precipitation, which is already only partly representative of the pattern over the integration time of the $^{10}$Be erosion signal, is probably even less so when considering the timescales over which these profiles develop. This particularly true in the SC range, where the profiles have evolved the time range covering a wet phase and then a dry phase. So it seemed better to take the simplest possible approach in this case.

*3. Choice of a concavity of 0.5 for the calculation of chi.*
Likewise, it was not our intention to really determining the best, or most relevant concavity values. We empirically observed that the segmentation was robust over the range over which the successive segments were reasonably linear (0.4-0.6), so the choice of 0.45 or 0.50 does not affect it. We simplified this part to clarify our intention.

*4. Alos DEM:* that's correct, it is the 30 m ALOS DEM, which is indeed, distinct from the Guatemalan national DEM.

*5. Figure 9*. We expanded the caption to make it a bit more self-explanatory

*6. Precipitation vs slope in controlling erosion rates*. We have reshuffled a bit this part, in order to better stress the fact that the most important point is that the slope-precipitation relationship intersect is close to 0, whereas the slope-erosion relationship predicts an absence of erosion on slopes shallower than 19°.

*7. Stressing the contribution of decreased rock uplift to the decrease in erosion earlier in the text*. We added a brief introductory paragraph to the section 5.2, explaining that in section 5.2 we successively review the contributions of climate to hillslope erosion and of tectonics to river incision (contribution of fast uplift in the AC range, river lengthening above the Polochic fault. We split the last part into an autonomous section 5.3, in order to better highlight the contribution of aridification.

*8. Stress why we think the dominant factor is precipitation*: we added some lines here to remind the results from previous sections that support the conceptual model.